# Response of simulated burned area to historical changes in environmental and anthropogenic factors: A comparison of seven fire models

Lina Teckentrup[1], Sandy P. Harrison[2], Stijn Hantson[3], Angelika Heil[1], Joe R. Melton[4], Matthew Forrest[5], Fang Li[6], Chao Yue[7], Almut Arneth[3], Thomas Hickler[5], Stephen Sitch[8], and Gitta Lasslop[1,5]

[1]Max Planck Institute for Meteorology, 20146 Hamburg, Germany
[2]School of Archaeology, Geography and Environmental Sciences (SAGES), University of Reading, Whiteknights, Reading, UK
[3]Karlsruhe Institute of Technology, Institute of Meteorology and Climate Research, Atmospheric Environmental Research, 82467 Garmisch-Partenkirchen, Germany
[4]Climate Research Division, Environment Canada, Victoria, BC, V8W 2Y2, Canada
[5]Senckenberg Biodiversity and Climate Research Institute (BiK-F), 60325 Frankfurt am Main, Germany
[6]International Center for Climate and Environmental Sciences, Institute of Atmospheric Physics, Chinese Academy of Sciences
[7]Laboratoire des Sciences du Climat et de l'Environnement–Institute Pierre Simon Laplace, Commissariat à l'Énergie Atomique et aux Énergies Alternatives (CEA)–Centre National de la Recherche Scientifique (CNRS)–Université de Versailles Saint Quentin
[8]College of Life and Environmental Sciences, University of Exeter

**Correspondence:** Gitta Lasslop (gitta.lasslop@senckenberg.de)

**Abstract.** Understanding how fire regimes change over time is of major importance for understanding their future impact on the Earth system, including society. Large differences in simulated burned area between fire models show that there is substantial uncertainty associated with modelling global change impacts on fire regimes. We draw here on sensitivity simulations made by seven global dynamic vegetation models participating in the Fire Model Intercomparison Project (FireMIP) to understand how

differences in models translate into differences in fire regime projections. The sensitivity experiments isolate the impact of the individual drivers on simulated burned area, which are prescribed in the simulations. Specifically these drivers are atmospheric $CO_2$ concentration, population density, land-use change, lightning and climate.

The seven models capture spatial patterns in burned area. However, they show considerable differences in the burned area trends since 1921. We analyse the trajectories of differences between the sensitivity and reference simulation to improve our

understanding of what drives the global trends in burned area. Where it is possible, we link the inter-model differences to model assumptions.

Overall, these analyses reveal that the largest uncertainties in simulating global historical burned area are related to the representation of anthropogenic ignitions and suppression and effects of land-use on vegetation and fire. In line with previous studies this highlights the need to improve our understanding and model representation of the relationship between human ac-

tivities and fire to improve our abilities to model fire within Earth system model applications. Only two models show a strong response to atmospheric $CO_2$ concentration. The effects of changes in atmospheric $CO_2$ concentration on fire are complex and

quantitative information of how fuel loads and flammability change due to this factor is missing. The response to lightning on global scale is low. The response of burned area to climate is spatially heterogeneous and has a strong interannual variation. Climate is therefore likely more important than the other factors for short term variations and extremes in burned area. This study provides a basis to understand the uncertainties in global fire modelling and the necessary improvements in process understanding and observational constraints to reduce uncertainties in modelling burned area trends.

*Copyright statement.* TEXT

# 1 Introduction

Wildfires are an important driver of vegetation distribution, and regulate ecosystem functioning, biodiversity and carbon storage over large parts of the world (Bond et al., 2005; Hantson et al., 2016a). Fire has strong impacts on climate through changing land surface properties, atmospheric chemistry and hence radiative forcing, as well as biogeochemical cycling (Bowman et al., 2009; Randerson et al., 2012; Ward et al., 2012; Yue et al., 2016; Li and Lawrence, 2017; Li et al., 2017; Lasslop et al., 2019). Estimates of the net effect of fire on the Earth system vary. Analyses based on observations of the pre-industrial period suggest that the contribution of fire to the overall climate–carbon-cycle feedback is substantial with $5.6 \pm 3.2$ ppm K-1 $CO_2$ (Harrison et al., 2018) while the strength of the global land climate–carbon-cycle feedback estimated from Earth system simulations (Arora et al., 2013) is 17.5 ppm K-1 (Harrison et al., 2018). However, comparing potential fire-induced losses from terrestrial carbon pools and stocks of solid pyrogenic carbon in soils and ocean, fire may also be a net sink of carbon and Earth system simulations show a negative effect of fire on radiative forcing (Lasslop et al., 2019). In addition to these consequences for the Earth System, wildfires directly impact society and economy (Gauthier et al., 2015) and human health can be seriously impaired (Johnston et al., 2012; Finlay et al., 2012).

Given the various impacts of fire on natural and human systems and the large uncertainties, it is important to improve the understanding on what controls the occurrence of wildfires and to know how fire regimes might change in the future.

Based on current process understanding the following drivers influenced burned area over the last decades to centuries:

Increasing atmospheric $CO_2$ concentration leads to increases in net primary production (Hickler et al., 2008) and decreased stomatal conductance reduces the plant transpiration and enhances water conservation in plants (Morison, 1985). It can lead to an increase in the abundance of woody plants ('woody thickening'; Wigley et al., 2010; Bond and Midgley, 2012; Buitenwerf et al., 2012) because $C_3$ plants are generally more competitive than $C_4$ plants under higher atmospheric $CO_2$ concentration (e.g. Ehleringer and Björkman, 1977; Ehleringer et al., 1997; Wand et al., 2001; Sage and Kubien, 2007). The impact of these various changes on burned area is complex. Increased productivity can lead to increased fuel availability, which can lead to increased burned area in water- and fuel-limited regions (Kelley and Harrison, 2014). On the other hand, decreased stomatal conductance and lower transpiration can lead to enhanced water conservation in plants. This increases the moisture content of soil as well as vegetation moisture content and consequently live and dead fuel moisture contents, which decreases flamma-

bility and in consequence reduces burned area. Woody thickening can lead to a reduction in burned area through changing the nature of fuel loads (Kelley and Harrison, 2014).

There is still controversy about whether humans increase or decrease fire overall: Although there is broad agreement that humans suppress fires in regions with high population density, observational studies are less clear about what happens in areas of low population density and show both increases or decreases due to human activities (see for instance Marlon et al., 2008; Bowman et al., 2011; Marlon et al., 2013; Vannière et al., 2016; Andela et al., 2017; Balch et al., 2017). Studies of the co-variation between population density and number of fires have shown that increasing population density leads to an increase in the number of ignitions or in the number of individual fires until peaking at intermediate population densities and drop subsequently (Syphard et al., 2009; Archibald et al., 2010). Burned area can be expressed as the number of fires multiplied by their fire size. The increase in burned area due to changes in ignitions is expected to differ between regions with varying population density as the largest fires occur in unpopulated areas (Hantson et al., 2015a). Global analyses find that the net effect of population density is a decrease in burned area (Bistinas et al., 2014; Knorr et al., 2014), with high uncertainties for low population density if the method allows for non-monotonic relationships (Knorr et al., 2014). Regional analyses tend to confirm this, but positive relationships between burned area and population density have been shown, for instance, for the least disturbed areas in the USA (Parisien et al., 2016).

Fire was used to manage croplands in pre-industrial times (e.g. Dumond, 1961; Otto and Anderson, 1982; Johnston, 2003) and is still common practice mainly in non-industrialized areas (i.e. Sub-Saharan Africa, parts of South East Asia, Indonesia and Latin America; e.g. Conklin, 1961; Rasul and Thapa, 2003). However fires in agricultural areas are common all over the world (Korontzi et al., 2006). Global analyses indicate a decrease of burned area (Bistinas et al., 2014; Andela and van der Werf, 2014) and fire size (Hantson et al., 2015b) with increases in cropland fraction. Fires on pasturelands have been estimated to contribute over 40% of the global burned area (Rabin et al., 2015). Analyses of global datasets find an increase of burned area with increases in grazing land cover (Bistinas et al., 2014) but reduced burned area on intensely grazed areas (Andela et al., 2017). Despite these analyses, the severe data gaps limit our level of understanding on how humans use fire in land management (Erb et al., 2017).

Lightning is the main source of natural ignitions (Scott et al., 2014). It is connected to convective activity and is therefore expected to change with global warming (Krause et al., 2014). Most of the burned area in boreal regions results from a few large fires (Stocks et al., 2002); these large fires are frequently ignited by lightning (Peterson et al., 2010). Veraverbeke et al. (2017) have shown that lightning ignitions drive the interannual variability as well as the long-term trends of ignitions in boreal regions.

Climate influences burned area through weather conditions and through its influence on vegetation (Bistinas et al., 2014; Forkel et al., 2017). Weather conditions (precedent precipitation, temperature and wind speed) influence fuel drying, wind speed additionally affects the rate of fire spread (Harrison et al., 2010; Scott et al., 2014). Vegetation type and fuel load are driven by climate and both strongly influence fire occurrence (Chuvieco et al., 2008; Pettinari and Chuvieco, 2016). Fires are limited under dry conditions due to low vegetation productivity and therefore insufficient fuel, and under wet conditions because the fuel is too wet to burn. The highest burned areas are therefore found in areas with intermediate moisture conditions (Krawchuk

and Moritz, 2011). There is no obvious disagreement in literature about how specific climatic factors influence fire. However, the relative importance of each factor, e.g. weather vs. vegetation, is still uncertain and varies spatially (Forkel et al., 2017). Fire models are sensitive to the meteorological forcing, different forcing datasets already lead to large differences in simulated burned area (Rabin et al., 2017a; Lasslop et al., 2014). The importance of factors also varies between small and large scales. Wind speed is an obvious driver of fire spread on the local scale, but it is difficult to extract this influence on the spatial resolution of global models (Lasslop et al., 2015).

Fire-enabled vegetation models simulate fire regimes in response to the combination of individual forcings, including atmospheric $CO_2$ concentration, population density, land-use change, lightning and climate. Individual fire-enabled vegetation models have been shown to simulate observed global patterns of burned area and fire emissions reasonably well (Kloster et al., 2010; Prentice et al., 2011; Li et al., 2012; Lasslop et al., 2014; Yue et al., 2014), but there are large differences between models in terms of regional patterns, fire seasonality and interannual variability, historical trends (Kelley et al., 2013; Andela et al., 2017) and responses to individual factors (Kloster et al., 2010; Knorr et al., 2014, 2016; Lasslop and Kloster, 2017, 2015). The fire model intercomparison project (FireMIP, Hantson et al., 2016a; Rabin et al., 2017a) provides a systematic framework to consistently analyse and understand the causes of these differences and to relate them to differences in the treatment of key drivers of fire in individual models. The FireMIP project provides simulations for a systematic comparison of fire-model behaviour based on outputs of a large range of models with identical forcing inputs. In addition to a reference historical simulation, sensitivity simulations were conducted for individual forcings, specifically atmospheric $CO_2$ concentration, population density, land-use change, lightning and climate. A recent evaluation of the FireMIP models indicates that the relationship with climatic parameters is captured well by models, the response to human factors is captured by some models and the response to vegetation productivity or the allocation of carbon to fuels needs refinement for most models (Forkel et al., 2019a). Comparisons of the FireMIP historical simulations found differences in transient model behaviour in the 20th century (Andela et al., 2017; van Marle et al., 2017). The causes of the differences and the reasons why different models show different responses are not yet understood.

In this multi-model study we use the historical simulation to show the overall modelled response of burned area to changes in environmental and human factors. We then compare the sensitivity experiments of the five most commonly used driving factors to document how simulated burned area responds to the individual forcing factors and relate inter-model differences of the burned area response to differences in model assumptions or parametrisation. We finally suggest implications of our results for model development and application.

## 2  Methods

The baseline FireMIP experiment (SF1) is a transient simulation from 1700–2013, in which atmospheric $CO_2$ concentration, population density, land-use, lightning, and climate change through time according to prescribed datasets. The baseline and sensitivity simulations start from the end of a spin-up simulation with equilibrated carbon pools (see Rabin et al. (2017a) for details of the experimental protocol). The five sensitivity experiments (SF2) are designed to isolate differences in model

behaviour associated with individual forcing factors. The model inputs and setup are the same as in SF1, but one of the forcings is kept constant at the value used in the spin-up throughout the experiment (see tab. 1). Thus, for example, in SF2_CO2, population density, land-use, lightning and climate inputs change each year, but atmospheric $CO_2$ concentration is held constant at 277.33 ppm for the whole of the simulation. The resulting difference in burned area between the simulations is then a
5   combination of the changes in the forcing and the sensitivity of the model to that forcing factor. Not all models performed every sensitivity experiment due to limitations in model structure (see tab. 2). Detailed model descriptions can be found in the corresponding literature listed in table A1. Two of the models (CLASS–CTEM and CLM) started the simulations later than the others (1861 and 1850, respectively) and due to limitations in data availability the reference year of the forcings used in the spin-up varies (see tab. 1). We account for these differences in starting years between models and of the forcing factors
10   by limiting our analysis to the period where all factors are different from the ones used in the spin-up (after 1921). These differences still influence the absolute differences, we therefore quantify the strength of the impact through the slope of a regression line and do not interpret the offset.

**Table 1.** Overview over the sensitivity experiments conducted by FireMIP-models (Rabin et al., 2017a). Rptd indicates the forcing was repeated over the given years. SF2_CO2 stands for fixed atmospheric $CO_2$ concentration, SF2_FPO for fixed population density, SF2_FLA for fixed land-use, SF2_FLI for fixed lightning, and SF2_CLI for fixed climate.

| Driving factor | Sensitivity Experiments | | | | |
| --- | --- | --- | --- | --- | --- |
| | SF2_CO2 | SF2_FPO | SF2_FLA | SF2_FLI | SF2_CLI |
| $CO_2$ | 277.33 ppm | transient | transient | transient | transient |
| Population density (PD) | transient | Fixed Year 1 | transient | transient | transient |
| Land-use change (LUC) | transient | transient | Fixed Year 1 | transient | transient |
| Lightning | transient | transient | transient | Rptd: 1901–1920 | transient |
| Climate | transient | transient | transient | transient | Rptd: 1901–1920 |

**Table 2.** Sensitivity experiments conducted by FireMIP models.

| Model | Sensitivity Experiments | | | | |
|---|---|---|---|---|---|
| | SF2_CO2 | SF2_FPO | SF2_FLA | SF2_FLI | SF2_CLI |
| CLASS–CTEM | x | x | x | x | x |
| CLM | x | x | x | x | x |
| INFERNO | x | x | x | | |
| JSBACH–SPITFIRE | x | x | x | x | x |
| LPJ–GUESS–SIMFIRE–BLAZE | x | x | x | | x |
| LPJ–GUESS–SPITFIRE | x | x | x | x | |
| ORCHIDEE–SPITFIRE | x | x | x | x | x |

## 2.1 Data processing and analysis of simulation results

Our analyses of the SF1 and SF2 simulations focus on the simulation of burned area but are complemented by effects on vegetation carbon pools for the SF2_CO2 simulation. We focus on the time series of global burned area over the historical simulation and the spatial patterns of differences in burned area between 1921 and 2013, as in this period all forcings are transient and different from the values used in the spin-up. Annual global values are an area weighted average using the grid cell area. We quantify the response of the models to each driving factor using the absolute difference in burned area between the baseline and the respective sensitivity experiment (SF1-SF2_i, with i in CO2, FPO, FLA, FLI, CLI). Positive differences mean that the transient change of the factor lead to an increase in burned area. We use the climate data operators (CDO version 2018: Climate Data Operators. Available at: http://www.mpimet.mpg.de/cdo) to process and remap the simulated outputs. We test the difference time series for trends over the period from 1921 to 2013 using the Mann-Kendall test, implemented in the R package Kendall (McLeod, 2011). We quantify the global trend as the slope of a linear regression and summarize the spatial distribution of trends by quantifying the area with significant positive trends and the area with significant negative trends. Due to a postprocessing error, INFERNO lacks two years in SF2_CO2 (2001 and 2002).

## 2.2 Model-data comparison

To evaluate the simulations of burned area, we compare the simulated burned area with remote sensing data products. Global burned area observations from satellites still suffer from substantial uncertainty, as reflected by the considerable differences in spatial and temporal patterns between different data products (Humber et al., 2018; Hantson et al., 2016a; Chuvieco et al., 2018; van der Werf et al., 2017). Using multiple satellite products in model benchmarking is one approach to take into account these observational uncertainties (Rabin et al., 2017a). In this study, we use three satellite products: GFED4 (Giglio et al., 2013), GFED4s (van der Werf et al., 2017) and FireCCI50 (Chuvieco et al., 2018). GFED4 is a gridded version of the MODIS Collection 5.1 MCD64 burned area product. It is known that this product strongly underestimates small fires, including cropland fires (e.g. Hall et al., 2016). In GFED4s, burned area due to small fires is estimated based on MODIS active fire (AF)

detections and added to GFED4 burned area. However, this methodology may introduce significant errors related to erroneous AF detections (Zhang et al., 2018). As a complementary product, FireCCI50 was developed using MODIS spectral bands with higher spatial resolution than MCD64. A higher resolution enhances the ability to detect smaller fires; however, this improvement is partially offset by suboptimal spectral properties of the bands. Both GFED4s and FireCCI50 have larger burned area than GFED4. Since all three products are based on MODIS data, the inter-product differences probably underestimate uncertainties associated with these products. A recent mapping of burned area for Africa using higher resolution Sentinel-2 observations indicates that all three products substantially underestimate burned area (Roteta et al., 2019). For the model evaluation we use temporally averaged burned area fraction for the years 2001–2013, the interval common to all three satellite products and the model simulations. We resample the model outputs to the lowest model resolution (CLASS-CTEM: 2.8125 x 2.8125°) with first order conservative remapping. We quantify the agreement between models and observations by providing the global burned area and the Pearson correlation coefficient for the between grid cell variation (see tab. 3). We choose the Pearson correlation as it quantifies the covariation of the spatial patterns, and is less sensitive to the highly uncertain absolute burned area values. Burned area has a strongly skewed distribution, with few high values and many small values close to, or equal to, zero. These few high values have a much higher contribution to the overall correlation (see figure A9 in Appendix) and therefore the metric is strongly determined by the performance of the model in areas with high burning. Square root or logarithmic transformation leads to more normally distributed values, that reduce this bias (see figure A9 in Appendix). As the logarithm transformation excludes grid cells with zero burned area, we adopt the square root transformation.

In spite of major advances in mapping burned area based on satellite data, these data products include major uncertainties. GFED4 and FireCCI50 provide uncertainty estimates for the burned area. Applying Gaussian error propagation, which assumes that errors are independent and normally distributed, yields uncertainty estimates of 0.01% (GFED4) and 0.2% (FireCCI50) of the global burned area, which is certainly an underestimation. The assumptions of normal distribution and independence are likely violated. The spread between global burned area data sets is probably a more realistic estimate. Since all the products rely on the MODIS sensor, this approach will not capture the full uncertainty. Nevertheless, to investigate the effect of data quality in the observations on the model-data comparison we use the burned area product uncertainty estimates (aggregated to model resolution assuming independence) to group the observations into points with low, medium and high uncertainty (low: within the 0–33rd percentile, medium: within the 33rd–66th percentile, and high: within the 66th–99th percentile of the relative uncertainty estimates = uncertainty / burned area). We then compute the correlations for data points with low, medium and high uncertainty separately.

## 3 Results and discussion

### 3.1 Simulated historical burned area

The models show magnitudes of annual global burned area between 354–531 Mha/yr for present day. This is comparable to the estimates obtained from the satellite products, which range from 345–480 Mha/yr (see fig. 1, tab. 3). The correlation coefficients between all of the simulations and the satellite observations are reasonable, with values ranging from 0.51 (CLASS–CTEM,

GFED4s) to 0.8 (ORCHIDEE–SPITFIRE, GFED4; see tab. 3). In general, the correlations with GFED4 are highest and with GFED4s lowest for almost all models - which may reflect the fact that most models do not explicitly simulate agricultural fires or may indicate inaccuracies in the mapping of agricultural fires in the GFED4s data set. The correlation coefficients strongly decrease with increasing observational relative uncertainty (see tab. A2). This shows that part of the mismatch in the spatial patterns between simulations and observations is a consequence of uncertainties in the satellite products themselves. The FireMIP models simulate the broad scale patterns in burned area reasonably well (see fig. A1), with maxima in the major fire-affected regions of the Sahel, southern Africa, northern Australia and the western USA. All of the models tend to overestimate the burned area in South America and also in the temperate regions of the USA. For a more detailed evaluation of the burned area see Forkel et al. (2019a).

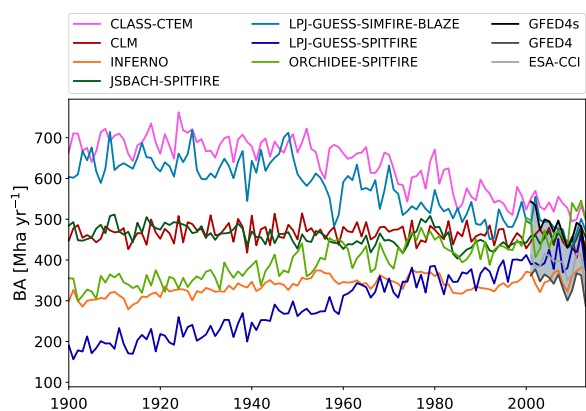

**Figure 1.** Annual global burned area (BA) in Mha yr$^{-1}$ for all FireMIP-models for 1921–2013 for the baseline experiment SF1. The shaded area indicates the range of annual global burned area values for the observations.

**Table 3.** Global burned area averaged over 2001–2013 in Mha yr-1 and the Pearson correlation coefficients between the baseline experiment SF1 for all FireMIP-models and the respective observation data. We use a square root transformation on both model and observations. All correlation coefficients are significant (p-value < 0.05).

| Model | Burned Area (Mha yr-1) | R(GFED4, model) | R(GFED4s, model) | R(FireCCI50, model) |
|---|---|---|---|---|
| CLASS–CTEM | 531 | 0.58 | 0.51 | 0.56 |
| CLM | 451 | 0.73 | 0.68 | 0.74 |
| INFERNO | 354 | 0.70 | 0.64 | 0.69 |
| JSBACH–SPITFIRE | 455 | 0.66 | 0.57 | 0.62 |
| LPJ–GUESS–SIMFIRE–BLAZE | 482 | 0.67 | 0.60 | 0.62 |
| LPJ–GUESS–SPITFIRE | 404 | 0.55 | 0.56 | 0.59 |
| ORCHIDEE–SPITFIRE | 474 | 0.80 | 0.72 | 0.79 |
| GFED4 | 345 | | | |
| GFED4s | 480 | | | |
| FireCCI50 | 389 | | | |

The simulated trend in burned area in the historical simulation differs between the models (see fig. 1). All models have a significant trend over the time series from 1921–2013 (see tab. 4). Models that have a relatively high total burned area initially (LPJ–GUESS-SIMFIRE–BLAZE, CLASS–CTEM) show a decline in burned area over the 20th century. Most models that have a low burned area (INFERNO, ORCHIDEE–SPITFIRE, LPJ-GUESS-SPITFIRE) show an increasing trend. JSBACH–

SPITFIRE and CLM have intermediate levels in burned area and show a weak decreasing trend over the 20th century.
Satellite records show a decline in global burned area since 1996 (Andela et al., 2017). However, as Forkel et al. (2019b) have shown, the significance of the observed global decline is strongly affected by the length of the sampled interval because of the high interannual variability in burned area and trends between products show only a low correlation (Forkel et al., 2019b).
No observations document the longer term trends in burned area. Charcoal records (Marlon et al., 2008, 2016) and carbon

monoxide data from ice-core records (Wang et al., 2010) are a proxy for biomass burning and show a global decrease in biomass burning over most of the 20th century. However, the charcoal records show an increase in burning since 2000 CE, but this discrepancy might reflect regional undersampling (for instance in Africa) or taphonomic issues of the charcoal record. A recent fire emission dataset (van Marle et al., 2017) merges information from satellites, charcoal records, airport visibility records and if no other information was available uses simulation results of the FireMIP models. This dataset is not included

to evaluate the models here as it is partly based on the simulations of the FireMIP models and as it provides only estimates for emissions not burned area.
The understanding of the drivers on simulated trends that we give below provides insights on what causes the simulated trends and which assumptions control the trend. These insights will help to understand which observational constraints and process understanding is required to improve global fire models.

**Table 4.** Trends (slope and standard error of a linear regression, [Mha yr$^{-1}$]) in annual global burned area for the years 1921-2013 for the baseline experiment SF1 and absolute difference time series of annual burned area. The trends for the forcing data sets are based on the the relative difference between the transient forcing and year 1920 value for SF2_CO2, SF2_FPO and SF2_FLA and the relative difference between the transient and the recycled forcing for SF2_FLI and SF2_FCL for the years 1921-2013 [%] (see tab. 1). Bold values indicate significance based on a Mann-Kendall test (p-value < 0.05). Experiments that are not available for specific models are indicated with n.a..

| Model | Sensitivity Experiments | | | | | |
|---|---|---|---|---|---|---|
| | SF1 | SF2_CO2 | SF2_FPO | SF2_FLA | SF2_FLI | SF2_CLI |
| CLASS–CTEM | **-2.238** | **-0.059** | **-0.754** | **-0.922** | 0.000 | 0.072 |
| | ± 0.116 | ± 0.008 | ± 0.052 | ± 0.049 | ± 0.001 | ± 0.134 |
| CLM | **-0.277** | **0.065** | **-1.05** | -0.065 | -0.048 | 0.046 |
| | ± 0.083 | ± 0.018 | ± 0.044 | ± 0.027 | ± 0.023 | ± 0.05 |
| INFERNO | **0.256** | **0.118** | **-0.571** | **0.303** | n.a. | n.a. |
| | ± 0.063 | ± 0.007 | ± 0.031 | ± 0.01 | | |
| JSBACH–SPITFIRE | **-0.304** | **0.574** | **-0.182** | **-0.873** | **-0.074** | 0.097 |
| | ± 0.077 | ± 0.020 | ± 0.038 | ± 0.051 | ±0.014 | ± 0.099 |
| LPJ–GUESS–SIMFIRE–BLAZE | **-2.161** | **-0.145** | **-0.847** | **-1.485** | n.a. | 0.249 |
| | ± 0.138 | ± 0.016 | ± 0.047 | ± 0.067 | | ± 0.144 |
| LPJ–GUESS–SPITFIRE | **2.351** | **0.986** | **1.345** | **1.845** | **0.015** | n.a. |
| | ± 0.087 | ± 0.032 | ± 0.050 | ± 0.044 | ± 0.006 | |
| ORCHIDEE–SPITFIRE | **1.383** | **0.035** | **0.520** | **0.859** | **0.334** | 0.033 |
| | ± 0.113 | ± 0.026 | ± 0.022 | ± 0.036 | ± 0.072 | ± 0.120 |
| | CO$_2$ | Population density | Land cover | Lightning | Temperature | |
| Forcing | **0.946** | **13.868** | **0.903** | **0.219** | **0.086** | |
| | ± 0.033 | ± 1.363 | ± 0.033 | ± 0.037 | ± 0.009 | |
| | | | | | Wind speed | |
| | | | | | **0.012** | |
| | | | | | ± 0.006 | |

## 3.2 Response of simulated burned area to individual drivers

The response of burned area to the individual factors is determined by the changes in the driving factors and the sensitivity of the model to these changes. The population density forcing dataset has the strongest trend in the relative differences between the transient forcing and the year 1920 value followed by the land-use and land cover change dataset. The trend in atmospheric CO$_2$ concentration is higher than the trend in the lightning dataset, which is more than twice as strong as in the air temperature.

Wind speed shows the lowest trend of all investigated driving factors (see tab. 4). Population density (SF2_FPO) and land-use change (SF2_FLA) cause the largest divergence between models in trends of burned area (slope between -1.05 and 1.345 Mha year$^{-1}$ and between -1.485 and 1.845 Mha year$^{-1}$, respectively). All models have a statistically significant trend in burned area for SF2_FPO as well as for SF2_FLA, except for CLM for SF2_FLA (see tab. 4, fig. 2 b and c). For SF2_CO2 all models have a significant trend, however, the magnitude of the trend is much smaller compared to the trend due to anthropogenic factors. LPJ–GUESS–SPITFIRE and JSBACH–SPITFIRE have strong trends (> 0.5 Mha year$^{-1}$), for all other models the slope is close to zero (< 0.15 Mha year$^{-1}$; see tab. 4, fig. 2 a). The differences between models are increasing over the 20th century for these first three experiments. The response to changes in lightning and climate generally shows much smaller trends but high inter-annual variability: none of the models has a significant trend for climate. Three models show significant (but inconsistent 0.014, 0.334 and -0.074 Mha year$^{-1}$) trends for lightning (see tab. 4). The interannual variability is stronger for climate. The mean standard deviation of the absolute differences averaged over all models is 30 Mha for climate and 7 Mha for lightning (only 3 Mha if the model with the strongest response is excluded; see fig. 2 d and e).

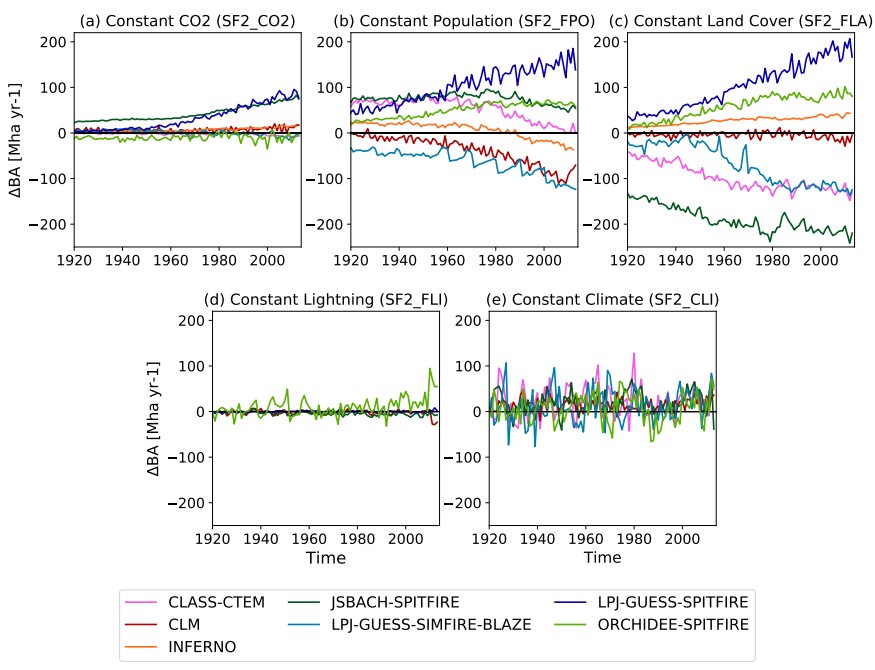

**Figure 2.** Absolute difference in annual global burned area ($\Delta$BA) in Mha across 1921 to 2013 between the baseline experiment SF1 and and the sensitivity experiments SF2_CO2 (a), SF2_FPO (b), SF2_FLA (c), SF2_FLI (d) and (e) SF2_CLI, where the specific forcing factors were set to the values used during the spin-up simulation (see tab. 1).

The spatial patterns of trends in burned area are mostly heterogeneous (see supplement figures A3–A7). The global trend can be dominated by changes in limited areas of the world, while the lack of a global trend can reflect opposing trends in different

regions. A detailed regional analysis is beyond the scope of this study, but we provide an alternative global view by quantifying the area affected by positive or negative trends (see fig. 3). This comparison shows that for most models larger areas show significant positive trends for the reference simulation (5 models), increasing atmospheric $CO_2$ concentration (5 models) and varying climate (5 models and 1 equal areas). There is no clear signal of either positive or negative trends across the models for the other simulations. For climate and lightning smaller areas have significant trends (see fig. 3). For ORCHIDEE–SPITFIRE and LPJ–GUESS–SPITFIRE all factors but climate cause a significant positive trend globally (see tab. 4) and larger areas have positive trends for all factors, except lightning for LPJ–GUESS–SPITFIRE (see fig. 3). On the other end of the model range LPJ–GUESS–SIMFIRE–BLAZE only shows a positive global trend for climate and atmospheric $CO_2$ concentration induced positive trends in larger areas than negative trends (see fig. 3).

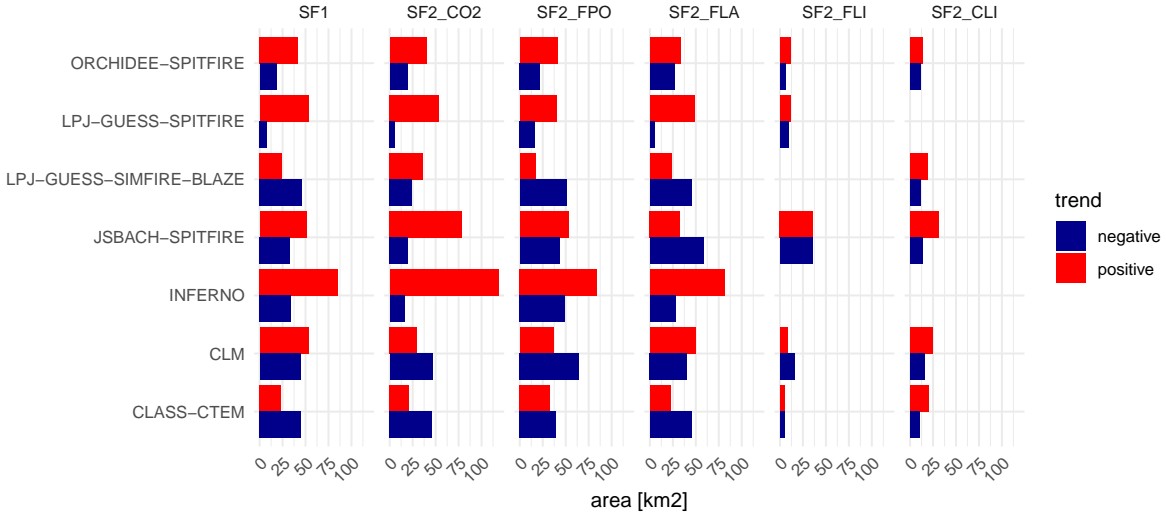

**Figure 3.** Area with a significant positive trend (red bar) or with a significant (Mann-Kendall test p<0.05) negative change (blue bar) in burned area fraction averaged over 1921–2013 for the baseline experiment SF1 and for the absolute differences in burned area fraction between the sensitivity experiments SF2 and SF1 (see tab. 1). Compare fig. A2 - A7.

In the following paragraphs we detail the inter-model differences and their causes for each sensitivity experiment.

### 3.2.1 Response of simulated burned area to atmospheric $CO_2$ concentration

The overall changes in burned area in individual simulations as a result of atmospheric $CO_2$ concentration changes are a complex response to multiple changes in vegetation: changes in land cover, fuel load, fuel characteristics and fuel moisture. Burned area can either increase due to higher availability of fuel loads or decrease due to changes in flammability caused by different fuel properties. The FireMIP-models react to increasing atmospheric $CO_2$ concentration in different ways: some models (JSBACH–SPITFIRE and LPJ–GUESS–SPITFIRE) show a strong increase in burned area, some (CLM and INFERNO) show a moderate increase, CLASS–CTEM shows a slight decrease, and LPJ–GUESS–SIMFIRE–BLAZE and ORCHIDEE–

SPITFIRE show a non-monotonic response (see fig. 2, a)). For all models, the trends over the 20th century are significant (see tab. 4).

We use changes in vegetation carbon to understand changes in fuel load and composition because information on the amount of fuel used within the fire models was not available for individual plant functional types (PFTs). All models show an increase in total vegetation biomass ('total', solid lines; see fig. 4), as expected because of higher productivity (Farquhar et al., 1980; Hickler et al., 2008) and increased water use efficiency (De Kauwe et al., 2013). The response of specific types of vegetation carbon to increasing atmospheric $CO_2$ concentration varies between the vegetation models. The biomass of $C_3$ vegetation (trees and $C_3$ grasses) increases in all of the models. The biomass of $C_4$ grasses increases in CLASS–CTEM, INFERNO, and JSBACH–SPITFIRE, but does not change in ORCHIDEE–SPITFIRE. Since ORCHIDEE–SPITFIRE was run with fixed vegetation distribution, changes in the extent of different PFTs can be ruled out as a cause of changes in vegetation carbon. There is a decrease in burned area in regions with abundant $C_4$ grasses (Sahel and North Australia) in this model, suggesting that changes in fuel type (increased $C_3$ tree biomass) results in changes in flammability in these regions. The carbon stored in $C_4$ grasses is reduced in response to increasing atmospheric $CO_2$ concentration in CLM and LPJ–GUESS–SIMFIRE–BLAZE and is fairly constant in LPJ–GUESS–SPITFIRE. This can be a result of a decrease in $C_4$ grass cover in LPJ–GUESS–SIMFIRE–BLAZE and LPJ–GUESS–SPITFIRE. However, since CLM was run with prescribed vegetation cover, the reduction in $C_4$ carbon must reflect the fact that any increase in $C_4$ grass biomass due to higher atmospheric $CO_2$ concentration is offset by greater losses through burning due to the increased total fuel load.

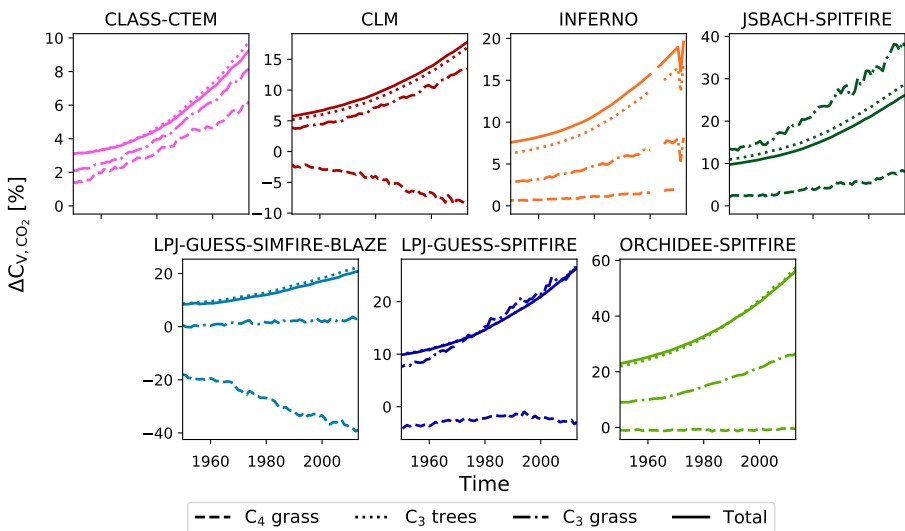

**Figure 4.** Relative difference in global carbon stored in $C_4$ grasses (dashed lines), in $C_3$ trees (dotted lines), in $C_3$ grasses (dash-dotted lines) and in total global carbon stored in vegetation (solid lines) between the baseline experiment SF1 and the sensitivity experiment SF2_CO2 (see tab. 1; $C_{V,CO_2}$) for 1950–2013 in % (annual averages). $C_4$ and $C_3$ grasses as well as $C_3$ trees only include natural PFTs (pastures and croplands excluded). Note that the y-axis limits differ between the panels. Due to a postprocessing error, INFERNO lacks two years (2001 and 2002).

CLM and LPJ–GUESS–SIMFIRE–BLAZE include an interactive nitrogen cycle, CLASS–CTEM a non-interactive nitrogen down-regulation. Effects of atmospheric $CO_2$ concentration on vegetation biomass for these three models are therefore at the lower end of the model ensemble. The strength of atmospheric $CO_2$ concentration effects on productivity is still uncertain and quantitative information about effects on fuel loads is not available. Comparisons with experimental data suggest that models
that do not include the nitrogen cycle overestimate the effect on productivity (Hickler et al., 2015). However, an analysis using an observation-based emergent constraint on the long-term sensitivity of land carbon storage shows that models from the Coupled Climate Model Intercomparison Project (CMIP5) ensemble that included an interactive nitrogen cycle underestimate the impact of atmospheric $CO_2$ concentration on productivity (Wenzel et al., 2016).

Soil moisture is used by several models to compute fuel moisture (see fig. 5). Soil moisture can be influenced by different
atmospheric $CO_2$ concentration as reductions in stomatal conductance can lead to increases in soil moisture, whereas increases in the leaf area index (LAI) caused by increased biomass of increased tree cover lead to higher transpiration and therefore lower soil moisture. Soil moisture increases slightly in four models (INFERNO, CLASS–CTEM, CLM, JSBACH–SPITFIRE), and decreases slightly in ORCHIDEE–SPITFIRE. Only LPJ–GUESS–SPITFIRE shows a strong decrease (5% in global average) in soil moisture (see fig. 6).

Models which include fuel load and moisture effects through threshold functions (see fig. 5, CLASS–CTEM, INFERNO, CLM) tend to show muted responses. Decreases in burned area appear to be largely caused by increases in soil moisture or tree cover. Increases associated with increasing fuel load are limited to regions with low biomass. The balance between these

effects differs between the models. CLASS-CTEM shows a small decrease in burned area globally, and the spatial pattern is dominated by areas with negative trends in burned area, but there are positive trends in dry regions (see fig. A3). The small global increase of burned area in INFERNO is likely related to increased fuel loads, negative trends in burned area only occur in the tropical regions (see fig. A3). INFERNO uses a constant burned area per PFT that is set to 0.6, 1.4 and 1.2

5   km$^2$ for trees, grass and shrubs, respectively. CLM shows increased global burned area, but increases are located in dry areas while the boreal regions show decreases. JSBACH–SPITFIRE and LPJ-GUESS–SPITFIRE respond to elevated atmospheric $CO_2$ concentration with a strong increase in burned area, likely driven by increases in fuel load. LPJ–GUESS–SPITFIRE additionally shows a strong decrease in soil moisture, which might explain why this model shows the strongest increase in burned area. ORCHIDEE–SPITFIRE shows lower burned area in response to elevated atmospheric $CO_2$ concentration but the

10   decreases are mainly localized in the regions with very high burned area (Sahel and Northern Australia; see fig. A3) and are likely driven by the increase in $C_3$ woody biomass (see fig. 4) as SPITFIRE is very sensitive to the type of fuel (Lasslop et al., 2014). LPJ–GUESS–SIMFIRE–BLAZE shows an initial increase and a decrease in burned area at the end of the simulation. The spatial pattern is mixed, the decrease in $C_4$ grass biomass indicates that woody thickening, either due to changes in land cover fraction or fuel composition is the reason for this reduction in burned area. An increase in woody plants with higher

15   atmospheric $CO_2$ concentration is expected (Wigley et al., 2010; Buitenwerf et al., 2012; Bond and Midgley, 2012). Their coarser and less flammable fuel can lead to reduced burned area. A recent study using an optimized empirical model indicates that increases in biomass led to decreases in burned area in regions with high fuel loads, likely due to increases in coarser fuels and increases in burned area in fuel limited regions (Forkel et al., 2019b).

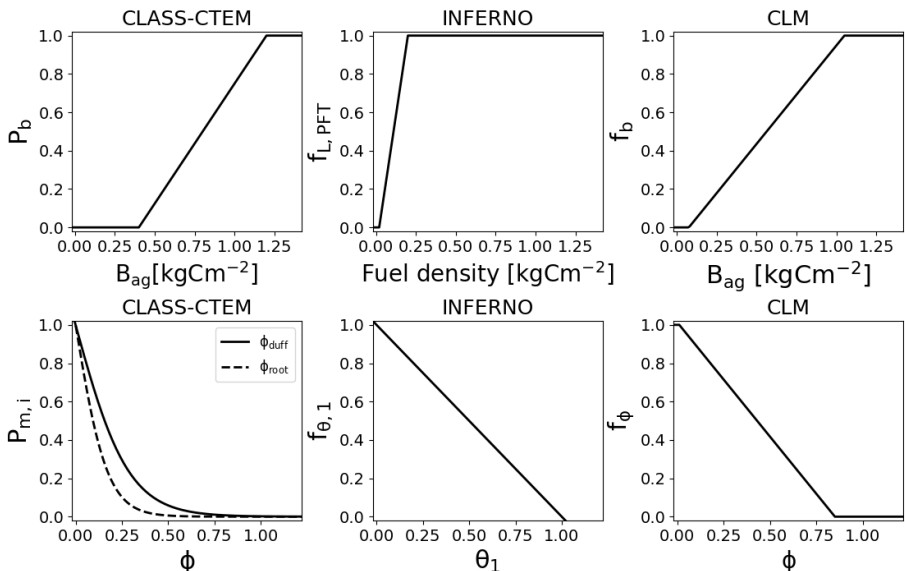

**Figure 5.** Impact of fuel load on the probability of fire ($P_b$) for CLASS-CTEM, on the fuel load index ($f_{L,PFT}$) for INFERNO and on fuel availability ($f_b$) for CLM (top panels). Impact of soil moisture content and soil wetness on fire for CLASS-CTEM, CLM, and INFERNO (bottom panels). In order to facilitate comparability, the soil moisture function for CLM is scaled to the value range [0,1].

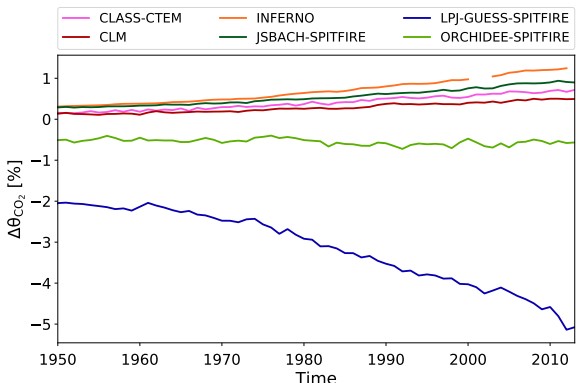

**Figure 6.** Annual average of the relative difference in volumetric soil moisture (CLM) and total soil moisture content (remaining models) between the baseline experiment SF1 and and the sensitivity experiment SF2_CO2 (see tab. 1; $\Delta\theta_{CO_2}$) for 1950–2013 in %. Due to a postprocessing error, INFERNO lacks two years (2001 and 2002).

### 3.2.2  Response of simulated burned area to population density

The population density forcing used for FireMIP increases in every region of the globe over time as well as in annual global values (Goldewijk et al., 2010). This increasing population density is associated with a monotonic increase of global burned area for LPJ–GUESS–SPITFIRE, and a monotonic decrease for LPJ–GUESS–SIMFIRE–BLAZE and CLM. The remaining
models show a peak in the impact of population density on burned area around 1950 and a subsequent decline (see fig. 2, b). Models however largely agree on a decreasing trend due to population density since 1921 (see tab. 4) and the ones that show a positive trend did not reproduce the relationship between population density and burned area in a multivariate model evaluation (Forkel et al., 2019a). Changes in population density therefore very likely contributed to a decrease in global burned area since 1921.
All the models, except LPJ–GUESS-SIMFIRE–BLAZE, include the number of anthropogenic ignitions ($I_A$) or the probability of fire due to anthropogenic ignitions ($P_{i,h}$ in CLASS–CTEM) in the calculation of burned area. Most of the models represent the number of anthropogenic ignitions with an increase up to a certain threshold number and then a decline, implicitly assuming that for high population densities humans suppress fires (SPITFIRE–models, INFERNO and CLM; see fig. 7). CLASS–CTEM, JSBACH–SPITFIRE and CLM include explicit terms to account for the effects of suppression not only on
ignitions but also on fire size, or duration, or both (see fig. 8). The combination of the ignition and suppression term in CLASS–CTEM leads to a maximum impact of humans on burned area at intermediate population density. The combination of ignition and suppression mechanisms dependant on population thresholds explains why most of the models have non-monotonic changes in burned area as population increases during the 20th century. LPJ–GUESS–SPITFIRE is the only model that shows a monotonic increase in burned area in response to increasing population density; other models that include the SPITFIRE fire
module (JSBACH, ORCHIDEE) show the non-monotonic trajectory that results from the shift from the dominance of ignitions

to that of suppression on burned area. ORCHIDEE–SPITFIRE has a much lower contribution from anthropogenic ignitions than LPJ–GUESS–SPITFIRE and therefore different spatial patterns of burned area (see fig. A1); JSBACH–SPITFIRE has an additional suppression term based on fire size data (Hantson et al., 2015a). The inclusion of additional suppression mechanisms may also explain the behavior of CLM, which shows a monotonic decrease in burned area over the 20th century.

LPJ–GUESS–SIMFIRE–BLAZE does not include anthropogenic ignitions explicitly but rather treats the net effect of changes in population density, which was optimized using burned-area satellite data (Knorr et al., 2014). This optimized net effect is a monotonic decrease of burned area with increases in population density. This explains why this model shows a monotonic decrease overall (see fig. A4).

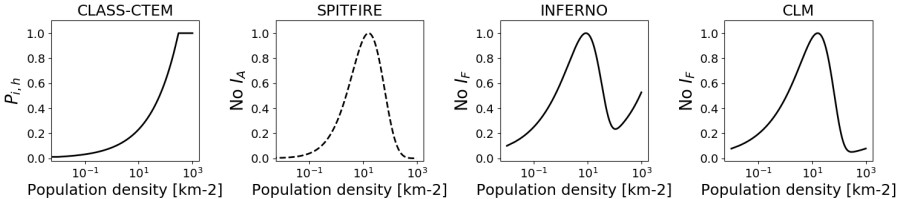

**Figure 7.** Variation in probability of fire due to human ignitions ($P_{i,h}$), anthropogenic ignitions (No $I_A$) or number of fires (No $I_F$) for changes in population density. Since all models use different units, the values are scaled to the value range [0,1].

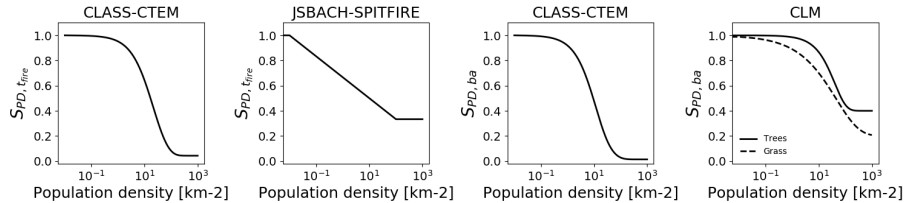

**Figure 8.** Suppression effects of population density on fire duration ($S_{PD,t_{fire}}$) for CLASS-CTEM and JSBACH SPITFIRE and suppression effects on fire size ($S_{PD,ba}$) for CLASS-CTEM and CLM. All models are scaled to the value range [0,1].

The models all agree that at high population density fire is suppressed. This leads to similarities in the spatial patterns of the
effect of population changes (see fig. A4) but they differ in their assumptions for low population density, the threshold where humans start to suppress fire and whether explicit suppression is included. The net or emerging effect of humans on burned area in models, however, also depends on the presence of lightning ignitions. The presence of lightning ignitions reduces the limiting effect of a lack of human ignitions on burned area. For the CLASS-CTEM model as soon as lightning ignitions are present, the net effect of humans is to suppress fires, even though the underlying relationship assumes an increase in ignitions
with population density (Arora and Melton, 2018, supplement). This may explain why global models assuming an increase of ignitions with increases in population density are able to capture the burned area variation along population density gradients (Lasslop and Kloster, 2017; Arora and Melton, 2018) and why global statistical analyses find a net human suppression also for low population density (Bistinas et al., 2014).

### 3.2.3 Response of simulated burned area to land-use change

The land-use change imposed in SF2_FLA is characterized by a strong decrease in forested areas, and an increase in pastures and croplands (Hurtt et al., 2011). The FireMIP-models do not show a uniform response of burned area to land-use change. LPJ–GUESS–SPITFIRE shows the strongest reaction with a monotonic increase in burned area with land-use change. INFERNO and ORCHIDEE–SPITFIRE also show an increasing trend, but of lower magnitude. CLASS–CTEM, JSBACH–SPITFIRE and LPJ–GUESS–SIMFIRE–BLAZE show a decreased burned area due to increased land-use. CLM also shows a decrease in burned area but this change is not significant (see fig. 2, c).

The FireMIP-models handle land-cover dynamics, the expansion of agricultural areas and fire in agricultural areas differently. Some of the models (CLASS–CTEM, CLM, JSBACH–SPITFIRE, ORCHIDEE–SPITFIRE) prescribe the vegetation distribution, so that the land cover fraction for all PFTs does not change through time in SF2_FLA while in the SF1 simulation the cover fractions of natural PFTs are reduced according to the expansion of agricultural areas. The other models simulate the distribution of the natural vegetation dynamically, but prescribe the agricultural areas. All models decrease the tree cover to represent the expansion of croplands over time. Land conversion due to the expansion of pasture is not represented in CLASS–CTEM. Only CLM includes cropland fires, INFERNO treats croplands as natural grasslands and all the other models exclude croplands from burning (see tab. 5). Therefore for all models except CLM and INFERNO, increases in cropland area lead to a reduction in burned area and the reasons for the divergence between the other models must be caused by the treatment of pastures.

**Table 5.** Treatment of agricultural fires (Rabin et al., 2017b). 'None' indicates the vegetation type does not burn or that deforestation fires are not represented in the model. The models treating pasture fire the same as grassland do not treat pasture as a specific PFT. The indication 'no pasture' means that there is no land cover change due to pastures.

| Model | Cropland fire | Pasture fire | Deforestation fire |
|---|---|---|---|
| CLASS-CTEM | None | no pasture | None |
| CLM | Yes | Same as grassland | Yes |
| INFERNO | Same as grasslands | Same as grassland | None |
| JSBACH-SPITFIRE | None | Higher fuel bulk density than grasslands | None |
| LPJ-GUESS-SIMFIRE-BLAZE | None | Harvest of biomass | None |
| LPJ-GUESS-SPITFIRE | None | Same as grassland | None |
| ORCHIDEE-SPITFIRE | None | Same as grassland | None |

In LPJ–GUESS–SIMFIRE–BLAZE pastures are harvested; this reduction in biomass leads to a decrease in burned area in addition to the decrease caused by exclusion of fire in croplands. In JSBACH–SPITFIRE, the expansion of pastures occurs preferentially at the expense of natural grassland and does not affect tree cover until all the natural grassland has been replaced (Reick et al., 2013). This assumption decreases the effect of land cover conversion on tree cover. Additionally, in JSBACH–SPITFIRE the fuel bulk density of pastures is higher than that of natural grass by a factor of two, which decreases fire spread

and thus burned area (Rabin et al., 2017b). This difference reduces burned area in pastures compared to natural grassland. In CLASS–CTEM, which also shows a decline, pastures are not included, the only land conversion is due to the expansion of croplands.

LPJ–GUESS–SPITFIRE and ORCHIDEE–SPITFIRE react with an increase in burned area to the expansion of land-use since they treat pastures as natural grasslands. The SPITFIRE fire module is very sensitive to the vegetation type with very high burned area for natural grasslands due to higher flammability compared to woody PFTs (Lasslop et al., 2014, 2016). Fuel bulk density is an important parameter but additionally grass fuels dry out faster leading to an increase in flammability and therefore burned area if forested areas are converted to grasslands. LPJ–GUESS–SPITFIRE computes the vegetation cover dynamically, so that an increase in burned area reduces the cover fraction of woody types, which might explain the stronger response compared to ORCHIDEE–SPITFIRE. In CLM, pastures are represented by increased grass cover. The biomass scaling function does not distinguish fuel types (see fig. 5), therefore the lower fuel amount of grasslands could lead to a decrease in fire probability, while the maximum fire spread rate depends on the vegetation type and is higher for grasslands (Rabin et al., 2017b). The inclusion of cropland and deforestation fires dampen the effect of land-cover change on global burned area. In INFERNO, agricultural regions are not defined explicitly. Instead, woody PFT types are excluded on agricultural area (Clark et al., 2011). INFERNO includes an average burned area for each PFT in the calculation of the burned area per PFT which leads directly to increasing grass cover resulting in higher burned area (Mangeon et al., 2016; Rabin et al., 2017b).

Land-use was already identified as a main reason for inter-model spread in the CMIP5 ensemble (Kloster and Lasslop, 2017). We show that this largely reflects the way pastures are treated, as most models used here (except CLM and INFERNO) simply exclude croplands from burning.

### 3.2.4 Response of simulated burned area to lightning

Most of the models show a low response of burned area to lightning (see fig. 2), although lightning rates increase by 20% over the simulation period – an increase that is much larger than the 3.3% change between pre-industrial times and the present estimated from a recent modelling study (Krause et al., 2014). ORCHIDEE–SPITFIRE shows an increase in burned area between 1940–1960 and towards the end of the simulation. In comparison to the other SPITFIRE-models the differences seem to be related to two points. Firstly, ORCHIDEE–SPITFIRE uses a 12 times higher factor to convert lightning strikes to actual ignitions and anthropogenic ignitions that are 100 times lower than for the other models (see Rabin et al., 2017b). Secondly, although a partitioning factor (SGFED) varies regionally, the per capita ignition frequency is constant; in JSBACH–SPITFIRE and LPJ–GUESS–SPITFIRE, the per capita ignition frequency varies regionally. This results in strong differences in the spatial patterns of burned area (see fig. A1). In consequence, the strength of regions contributing to the global burned area varies between the models; ORCHIDEE–SPITFIRE shows much more burning in the tropical and far less burning in the temperate region. Whether a lightning turns into a fire depends on the local conditions at the time of the lightning strike. Differences in the spatial distribution and timing of fires can therefore lead to different responses between models even if lightning is used in the same way within the model. Our results show that even a substantial increase (20%) in lightning has little influence on

simulated global burned area. This is consistent with (Krause et al., 2014) who found that the pre-industrial to present increase in lightning, although this increase is much smaller, had little impact on burned area.

### 3.2.5 Response of simulated burned area to climate

Simulated burned area in FireMIP responds to changes in climate with strong interannual variability but only weak trends in burned area (see fig. 2, e). Only three models show a statistically significant trend in the global burned area according to a Mann-Kendall test (CLM, LPJ–GUESS–SIMFIRE–BLAZE,ORCHIDEE–SPITFIRE; see tab. 4). However, in all models the area showing an increased burned area in response to climate is higher than the area with decreased burned area (see fig. 3). Agreement in spatial patterns of trends between the models is however low (see fig. A7).

The influence of climate on burned area is complex; it influences burned area through the meteorological conditions and through effects on vegetation conditions that influence fuel load and fuel characteristics (Scott et al., 2014). We therefore correlated for each grid cell changes in physical parameters (precipitation, temperature, wind speed and soil moisture) and vegetation parameters (litter, vegetation carbon and grass biomass) with changes in burned area. We find that the correlation between the individual parameters and burned area is low (see fig. A8). The absolute rank correlations are lower at the monthly scale than at the annual scale. However, at the monthly scale the number of grid cells showing significant correlations with physical parameters is higher than the number showing significant correlations with vegetation parameters, indicating that changes in physical parameters have more influence at shorter time scales than changes in vegetation parameters. This difference disappears with the aggregation to annual time scale. On the annual time scale, however, the mean absolute rank correlation is slightly higher for the vegetation parameters. Soil moisture which is also influenced by vegetation has a slightly higher correlation compared to precipitation, temperature and wind speed too. This indicates that vegetation parameters are more influential on the longer annual time step and physical parameters on the monthly time step. The relationship between precipitation or soil moisture and burned area is expected to be negative, while the impact of temperature is expected to be positive. This is clearly reflected in the percentage of positively significant correlations at the annual scale, but is less clear at the monthly time step. This might reflect that the seasonality of temperature, precipitation and vegetation parameters is often synchronized and therefore the effects of the parameters cannot be separated. The low correlation between individual parameters and burned area reflects the complex interactions between the climatic drivers, vegetation conditions and fire weather.

The impact of climate on the interannual variability, however, is strongly expressed in the simulated burned area. This is consistent with the finding that recent precipitation changes influence interannual variability in fire but have little impact on recent longer-term trends (Andela et al., 2017). To fully understand the impact of the changes in climate, a number of simulations would be necessary, where only individual climate parameters change while the others are kept constant. In addition, simulations where combinations of variables change, might give further insights on the synergies between the variables. An alternative approach, given the complex interactions between climate and vegetation parameters, might be to disentangle the model signals using multivariate analysis (see e.g. Forkel et al., 2019a; Lasslop et al., 2018).

### 3.3 Implications for model development and applications

Global vegetation models are an important tool for examining the impacts of climate change and are used in policy-relevant contexts (IPCC, 2014; Schellnhuber et al., 2014; IPBES, 2016). Given the various influences of fire on the ecosystems (Bond et al., 2005), the carbon cycle and climate (Lasslop et al., 2019), improvements of global fire models are particularly important.

The main concern for model applications is the large spread of the historical simulated burned area. It remains difficult to evaluate and optimize the transient burned area simulations as the period observed by satellites is still short and the trends are not robust (Forkel et al., 2019b). Fire proxies (charcoal and ice-cores) give information on biomass burning over longer time scales. They do not confirm the recent decrease in burned area detected by satellites, but also only contain very few datapoints for that period (Marlon et al., 2016). For a valid comparison with the long term fire proxies, including estimates of deforestation fires in the models will be crucial, as land-use change fire emissions likely have a strong contribution to the signal (Marlon et al., 2008). An improved understanding of uncertainties in observed trends of fire regimes is therefore necessary. Only robust information should be included in models.

Our analysis shows which parts of the models are particularly important to simulate changes in burned area and need additional observational constraints or improved process understanding. In line with previous research (Bistinas et al., 2014; Hantson et al., 2016a, b; Andela et al., 2017), the large divergence in the response to human activities between the FireMIP models shows that the human impact on fires is still insufficiently understood and therefore not constrained in current models. We identify land-use change as the major cause of inter-model spread. Only one model explicitly includes fires associated with land-use and land cover change (cropland and deforestation fires), all the other models only include such effects through changes in vegetation parameters and structure. The inclusion of cropland fires is certainly important to understand and project changes in emissions, air pollution and the carbon cycle (Li et al., 2018; Arora and Melton, 2018). Cropland fires are, due to their small extent and low intensity, still a major uncertainty in our current understanding of global burned area (Randerson et al., 2012). Biases in the spatial patterns of burned area and the relationship between cropland fraction and burned area can therefore be expected. High resolution remote sensing may help to improve the detection (Hall et al., 2016). Moreover, understanding why and when humans burn croplands on a regional scale may help to find an adequate representation of cropland fires within models and avoid overfitting to observational datasets. As croplands are simply excluded from burning in most models (except two), the spread of the other models is likely related to the treatment of pastures. Fires on pasturelands have been estimated to contribute over 40% of the global burned area (Rabin et al., 2015). Pasture fires are not treated explicitly in any of the models, although some models slightly modify the vegetation on pastures by harvesting or changing the fuel bulk density (see tab. 5). Expansion of pastures is mostly implemented by simply increasing the area of grasslands. Information on how fuel properties differ between pastures and natural grasslands could therefore help to improve model parametrisations. Prescribing fires on anthropogenic land covers can be a solution for certain applications of fire models (Rabin et al., 2018). Grazing intensity was found to be related to decreases in burned area (Andela et al., 2017). Models so far represent the area that is converted due to land cover change but not the intensity of land-use. This was partly due to the lack of global data regarding land use intensity which is now becoming available and provides new opportunities for fire model development (e.g.

the LUH2 dataset; Hurtt et al., 2017). In the sensitivity simulations shown here, even models that decrease burned area due to land-use and land cover change do not show a further decrease over the last decade. This indicates that model input datasets, explicit in time and space, for land-use intensity and grazing intensity are necessary for fire projections. The level of socioeconomic development also modifies the relationship between humans and burned area (Andela et al., 2017; Forkel et al., 2017).

Regional analysis of remote sensing data could be highly useful, as a global relationship between burned area and individual human factors as assumed in many models and also statistical analysis is not likely. Assumptions on how different human groups (hunter-gatherers, pastoralists, and farmers) use fire have been included in a paleofire model (Pfeiffer et al., 2013). The development of such an approach for modern times would be highly valuable for fire models that aim to model the recent decades and future. Deforestation fires are only included in one model (CLM). As deforestation fires are likely a strong source

of biomass burning over the longer time scales, accounting for deforestation fires will be crucial for a model comparison with the charcoal record.

We also find inter-model agreement for certain aspects. For instance, burned area is suppressed at high population densities, which leads to a similar spatial response to population density (see fig. A4). Moreover, most models show a reduction of the global burned area due to changes in population density. The response functions of burned area to population density of the

two models that increase burned area is less in line with response functions derived from global datasets (Forkel et al., 2019a). As a strong human suppressive effect is well supported by satellite observations (Andela et al., 2017; Hantson et al., 2015b), a reparametrisation of these responses would be reasonable.

We show that, although all models show an overall increase in biomass as a consequence of increasing atmospheric $CO_2$ concentration, models disagree about whether this results in an increase or decrease in burned area. The disagreement reflects the

complex ways in which changes in atmospheric $CO_2$ concentration influence vegetation properties, which results in different responses in different ecosystems. For LPJ-GUESS-SPITFIRE and JSBACH-SPITFIRE the $CO_2$ fertilization effect considerably contributed to an increase in burned area. Such an effect is so far only supported for fuel limited areas (Forkel et al., 2019b). The assumption that the influence of higher fuel load on burned area levels off for high fuel loads as used in other models could help to reduce this increase in burned area in regions with higher fuel load.

Climate and lightning have a much lower effect on the trends than the other factors. While this study focuses on the trends, research on the short term variability and extreme events will be highly useful to investigate fire risks. The influence of climate and lightning on fire are therefore important research topics even if we find a comparably low influence on the long term trends. Moreover the trends in climate parameters may increase for the future and therefore the influence on burned area might increase.

In contrast to many model simulations that use a lightning climatology based on satellite observations, the FireMIP experiments were driven by a transient dataset of lightning activity created by scaling a mean monthly climatology of lightning activity using convective available potential energy (CAPE) anomalies of a global numerical weather prediction model. Since climate changes can be expected to cause changes in lightning, it will be important to develop transient lightning datasets for climate change studies on fire. Using present day lightning patterns, for example, will certainly lead to an overestimation of

lightning strikes in regions with drier climate projected in the future. But not only spatial patterns of lightning are important,

the co-variation with climate as well as the temporal resolution of the input dataset determine the influence on burned area (Felsberg et al., 2018). Although we do not detect large signals in global burned area due to changes in lightning, lightning is known to be an important cause of ignitions regionally and is potentially involved in more complex interactions between fire, vegetation and climate, which can speed up the northward expansion of trees to the north in boreal regions (Veraverbeke

et al., 2017). Thus, although our results suggest that the influence of increasing lightning is negligible at a global scale, it is a potentially important factor for process-based models that aim to model interactions between fire, vegetation and climate.

Recent advances in remote sensing products have high potential to support model development. However, remotely sensed burned area datasets alone are not a sufficient basis to evaluate fire models as many model structures can lead to reasonable burned area patterns. The emergence of longer records of burned area and the increasing availability of information on other

aspects of the fire regime considerably improve opportunities to evaluate and improve our models. The FRY database (Laurent et al., 2018) and the global fire atlas (Andela et al., 2018), for example provide information on fire size, numbers of fire, rate of spread, and the characteristics of fire patches. These datasets will be useful to, for instance, separate effects of ignition and suppression. Rate of spread equations in global fire models are at present either very simple empirical representations tuned to improve burned area or based on laboratory experiments (Hantson et al., 2016a). The mentioned datasets now offer the

opportunity to derive parameters for rate of spread equations at the spatial scales these models operate on. Fire size and rate of spread are important target variables besides burned area that can determine the impacts of fire. The effects on vegetation (combustion of biomass and tree mortality; Williams et al., 1999; Wooster et al., 2005) and on the atmosphere (Veira et al., 2016) are a function of fire intensity, which is also included in the FRY database (Laurent et al., 2018). A better evaluation of such parameters can enhance the usability of fire model simulations.

The specific model application has a strong influence on judging the validity of a model. Our analyses of the controls on the variability of fire suggest that human activities drive the long term (decadal to centennial) trajectories, while considering climate variability may be sufficient for short-term projections. Changes in the trends of the driving factors may change this balance. For instance, stronger changes in climate into the future may increase the relative importance of climate for long term fire projections in the future.

**4 Summary and conclusions**

This comprehensive analysis of the influences of climate, lightning, atmospheric $CO_2$ concentration, population density and land-use and land cover change provides improved understanding of the relation between simulated historical trends in burned area and process representations in the models. It shows in detail which model responses of burned area to environmental factors can be understood, how these are related to the model equations, and how these translate into trends of burned area for

the historical period.

The analysis of the sensitivity experiments shows that: The increase in atmospheric $CO_2$ concentration over the 20th century leads to increased burned area in regions where fuel loads increase, but to decreased burned area in regions where tree density or coarse fuels with lower flammability increase or elevations in soil moisture decrease flammability. Although models agree

that the amount of available fuel increases, the type of fuel and vegetation composition are critical to understand the influence of atmospheric $CO_2$ concentration on simulated burned area.

Most models agree on a decrease in burned area due to increases in population density. Most models link the number of ignitions to population in a way that ignitions increase initially at low population densities. In densely populated regions, all models assume that the effect of anthropogenic ignitions is outweighed by fire suppression and the increased fragmentation of the landscape by anthropogenic land-use. It would be useful to develop an approach that represents local human-fire relationships, but this will likely remain a long term challenge and requires the synthesis of knowledge from various research fields.

The simulated response of burned area to land-use and land cover change depends on how fires in cropland and pastureland are treated in each model. Most models simply exclude croplands from the burnable area, therefore the treatment of pastures causes the largest part of the model spread. Models that do not allow fire in croplands, and either harvest biomass in pastures or assume specific vegetation parameters, show a reduction in burned area. Models that treat pastures as natural grasslands and distinguish different fuel types or strongly increase burned area for grasslands show an increase in burned area. Improved knowledge on the effects of land-use intensity on burned area and the development of appropriate forcing datasets could strongly support model development.

The models are comparatively insensitive to changes in lightning, likely because lightning ignitions are not a limiting factor in many regions with very high burning activity. Previous studies however show the importance of lightning and changes in lightning for burned area in the boreal region. Therefore especially regional studies should pay attention to this factor.

None of the models shows a strong trend due to changing climate but all of them show a strong influence of climate on the interannual variability. Climatic and ecosystem parameters are only able to explain a rather small part of this variation, with stronger correlations for the ecosystem parameters on the longer annual time scale and stronger relationship with climatic parameters on the monthly time scale.

Different drivers of burned area affect different time scales: the anthropogenic factors influence long term variability, while climate, and lightning affect short-term variability. Understanding the influence of climate and lightning is especially important for interannual variability and extreme events. On the other hand understanding the impact of anthropogenic drivers are likely more important for the longer term changes of fire as for instance needed in Earth system models. Changes in the trends of the forcing parameters might however affect the balance between them.

The uncertainties in global fire models need to be taken into account in model applications, for instance if model simulations are to be used to support climate adaptation strategies. Model ensemble simulations can give indications of such uncertainties. Therefore the results of this study provide a basis to interpret uncertainties in global fire modelling studies. The information content on the spatial variability of burned area has been well exploited in previous studies and models reproduce the spatial patterns in a reasonable way. The temporal information of the satellite data is increasing with the increasing length of the record and has a higher potential to contain new information to support the improvement and evaluation of global fire models. We here provide a summary of which model assumptions need additional constraints to efficiently reduce the uncertainty in temporal trends.

*Code availability.* TEXT

*Data availability.* Datasets will be available after acception of the paper

*Code and data availability.* TEXT

*Sample availability.* TEXT

# Appendix A

## A1

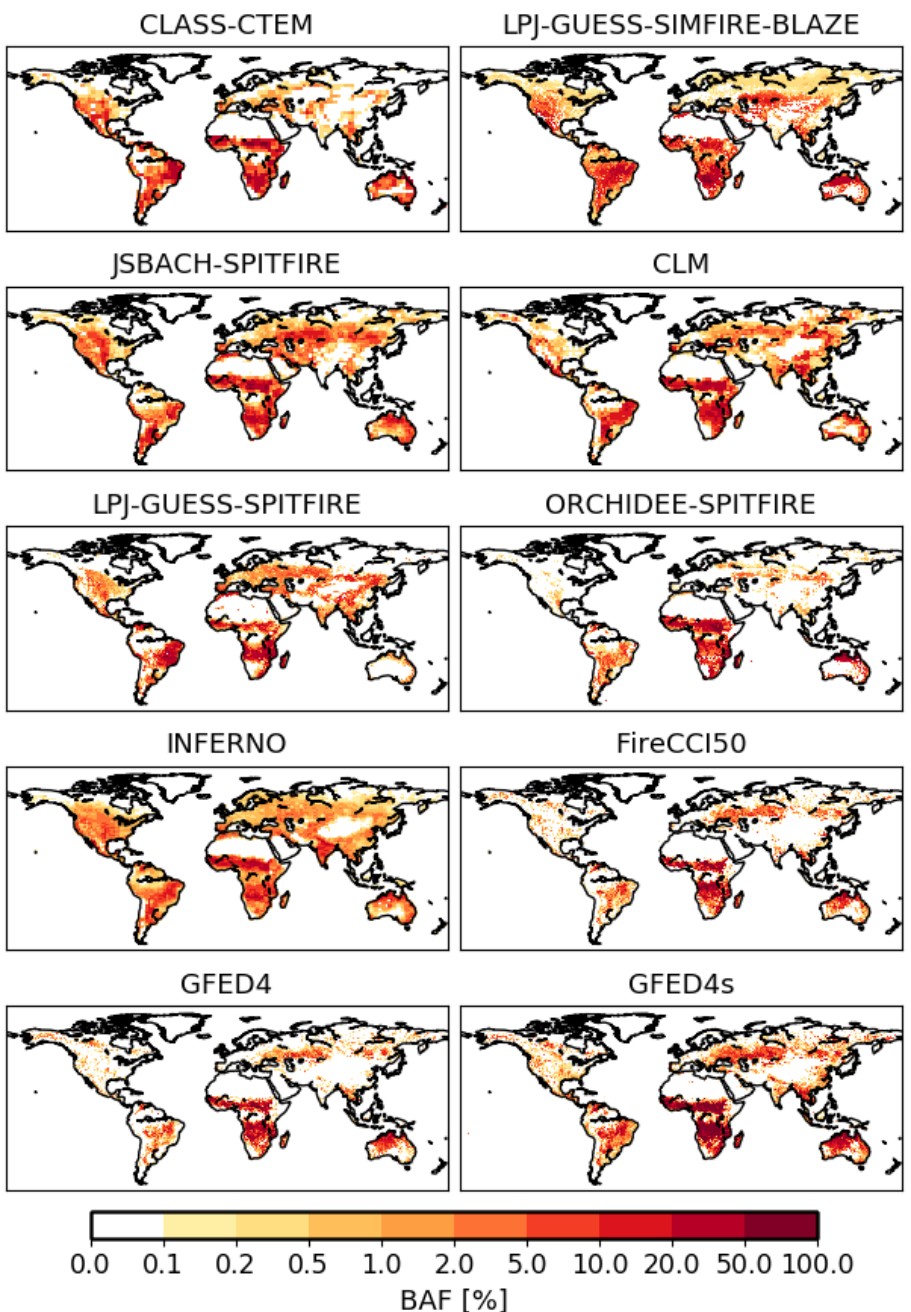

**Figure A1.** Spatial distribution of annual burned area fraction (BAF) in % for the baseline experiment SF1 and observation data, averaged over 2001-2013.

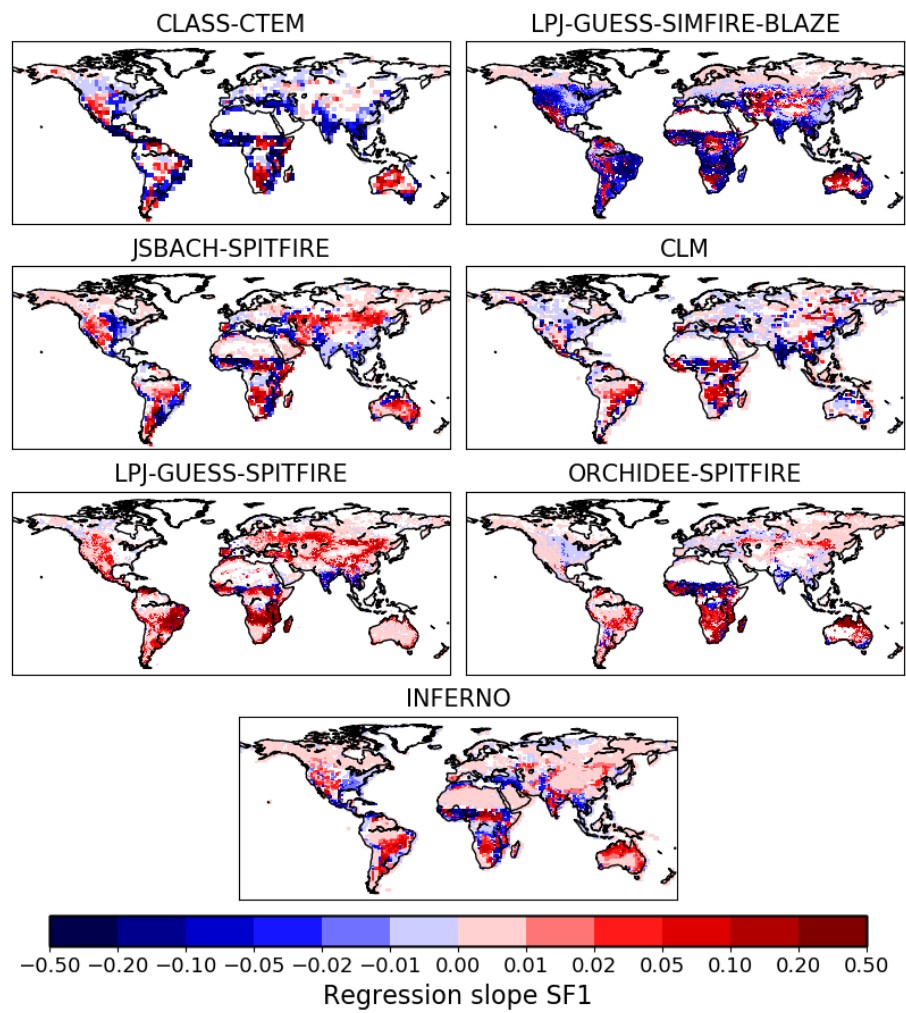

**Figure A2.** Spatial distribution of regression slopes for the baseline experiment SF1 over 1921-2013.

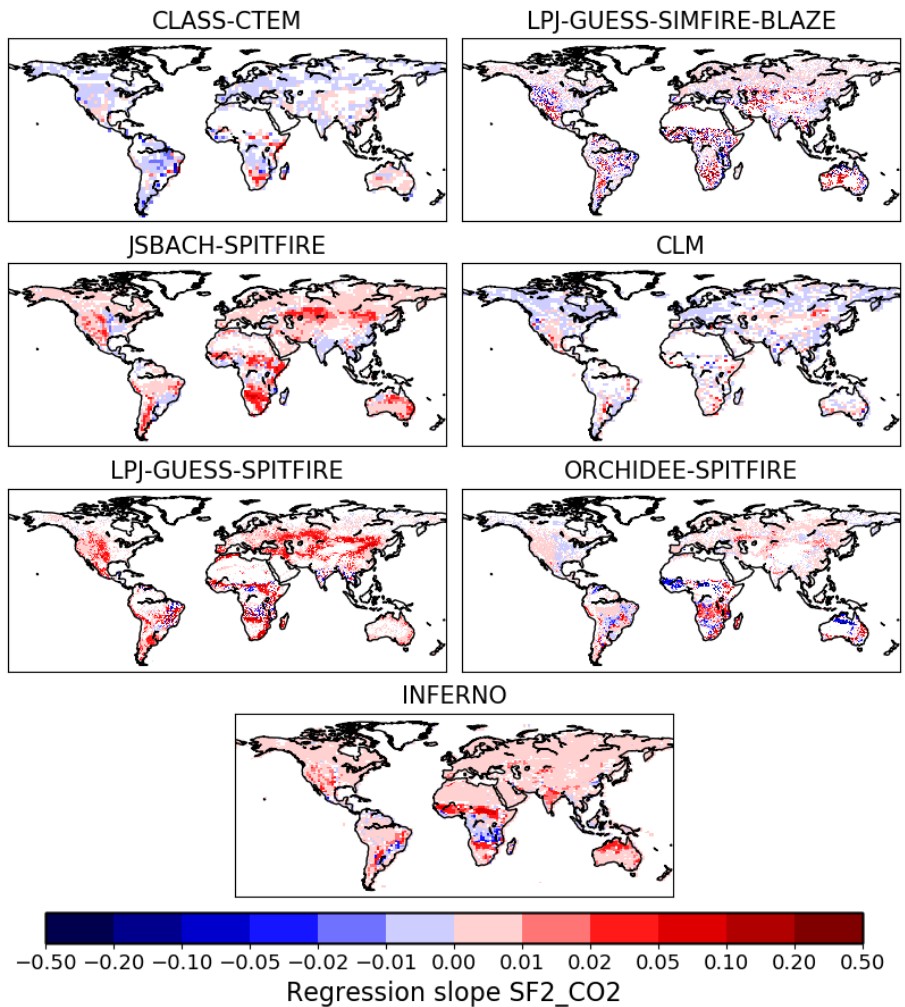

**Figure A3.** Spatial distribution of regression slopes for the difference between the baseline experiment SF1 and the sensitivity experiment SF2_CO2 (SF1–SF2_CO2; see tab. 1) over 1921-2013.

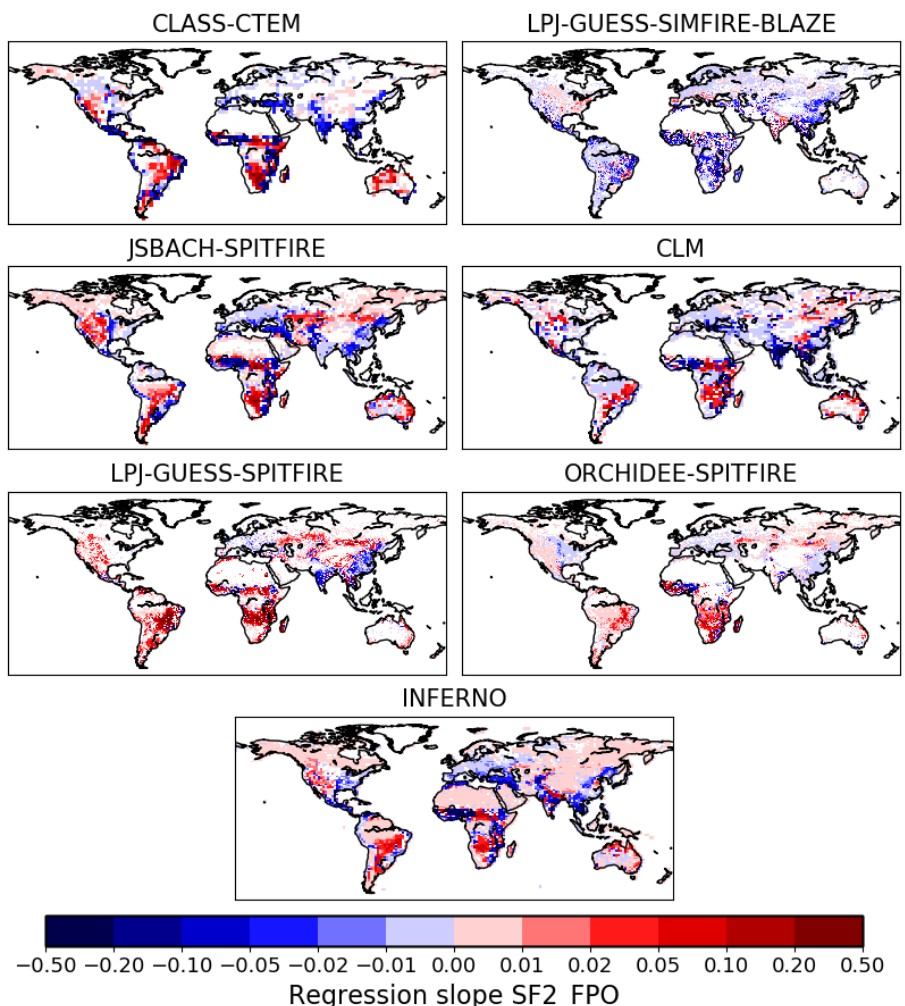

**Figure A4.** Spatial distribution of regression slopes for the difference between the baseline experiment SF1 and the sensitivity experiment SF2_FPO (SF1–SF2_FPO; see tab. 1) over 1921-2013.

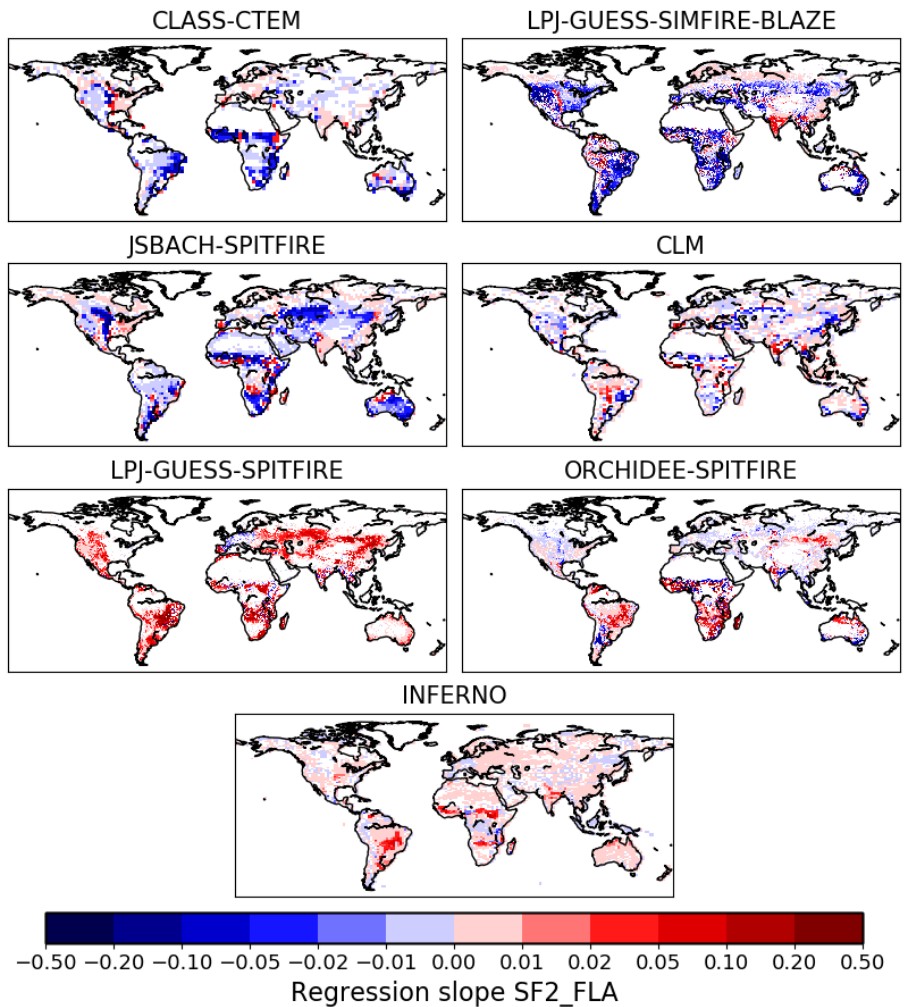

**Figure A5.** Spatial distribution of regression slopes for the difference between the baseline experiment SF1 and the sensitivity experiment SF2_FLA (SF1–SF2_FLA; see tab. 1) over 1921-2013.

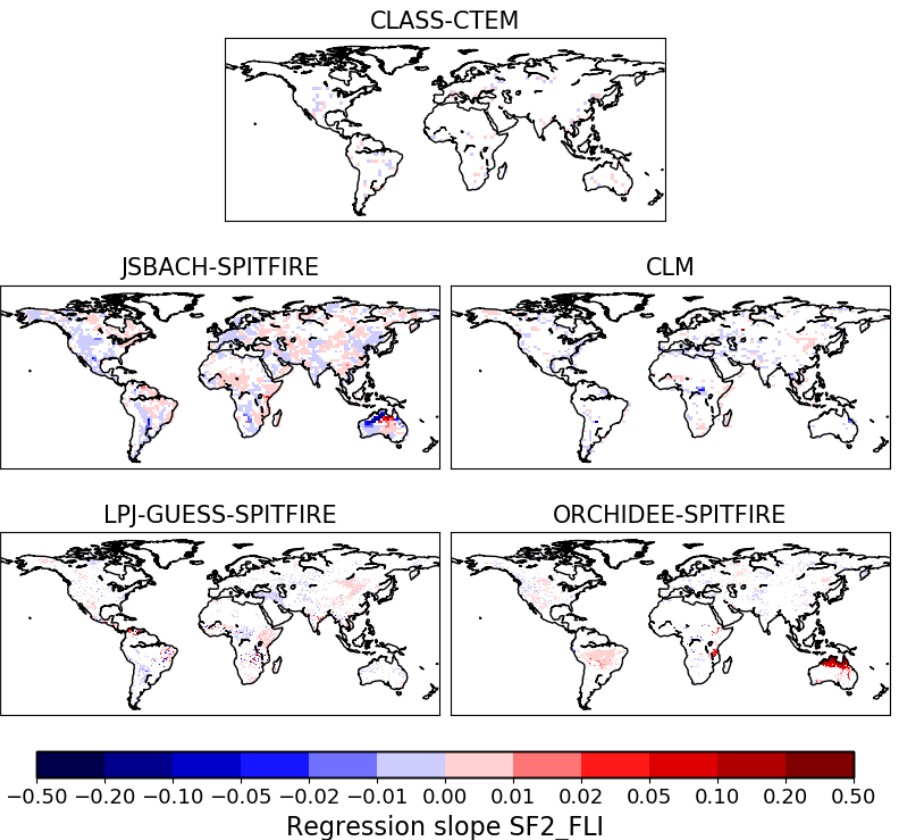

**Figure A6.** Spatial distribution or regression slopes for the difference between the baseline experiment SF1 and the sensitivity experiment SF2_FLI (SF1–SF2_FLI; see tab. 1) over 1921-2013.

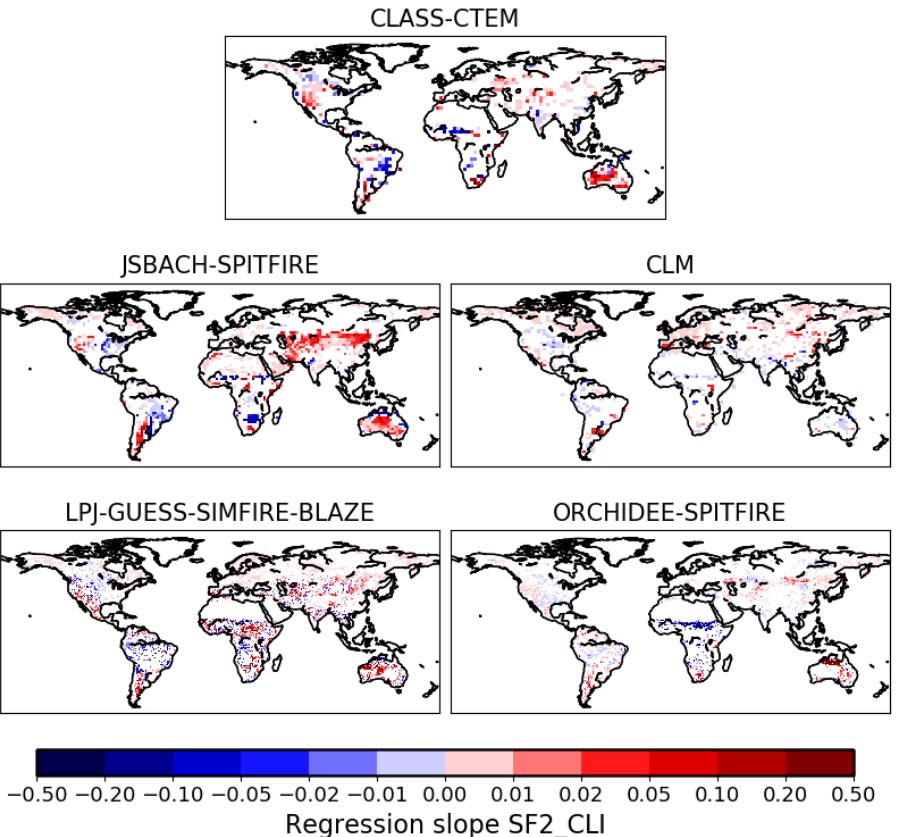

**Figure A7.** Spatial distribution of regression slopes for the difference between the baseline experiment SF1 and the sensitivity experiment SF2_CLI (SF1–SF2_CLI; see tab. 1) over 1921-2013.

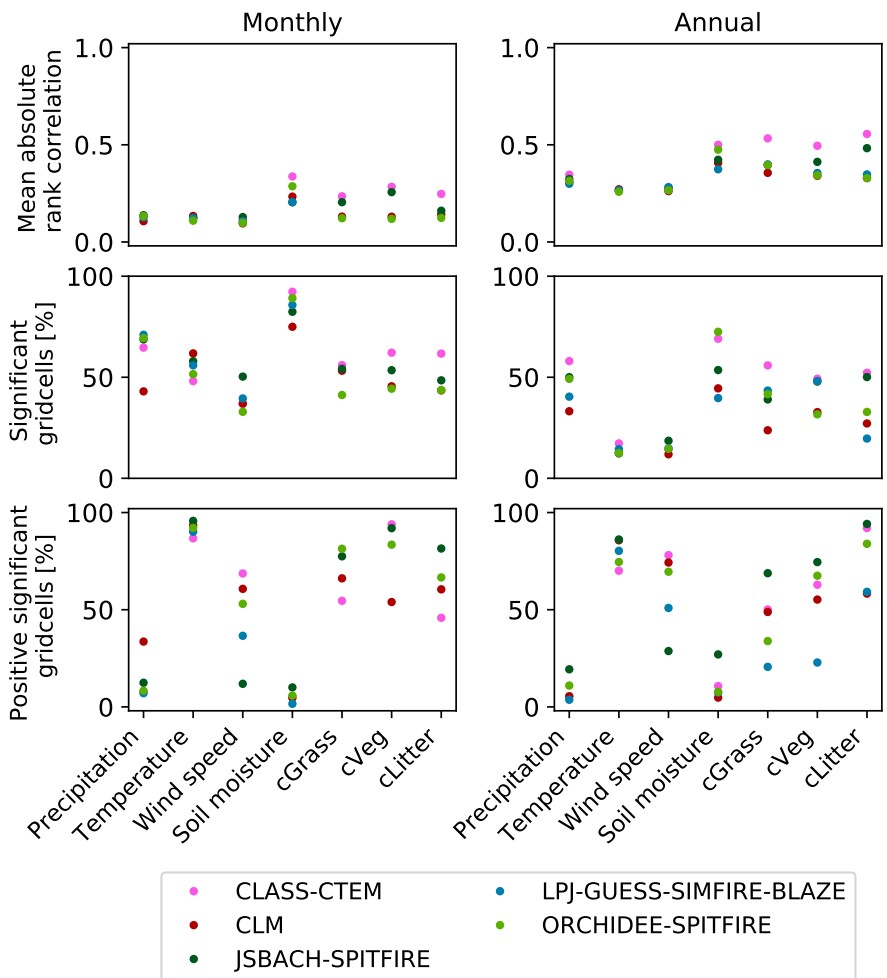

**Figure A8.** Spearman rank-order correlation coefficient for each grid cell over 1921–2013 between the difference between the baseline experiment SF1 and the sensitivity experiment SF2_CLI (see tab. 1) for annual burned area fraction and precipitation, temperature, wind speed, carbon stored in litter, carbon stored in vegetation, carbon stored in grass and in soil moisture, respectively. The upper panel shows the mean absolute rank correlation, i.e. the spatial average over the absolute and significant (p-value $< 0.05$) Spearman rank-order correlation coefficients where the relative difference in burned area fraction is $> 0.1$. The second panel shows the proportion of grid cells with a significant correlation. The lowest panels indicate the percentage of significant grid cells with a positive correlation.

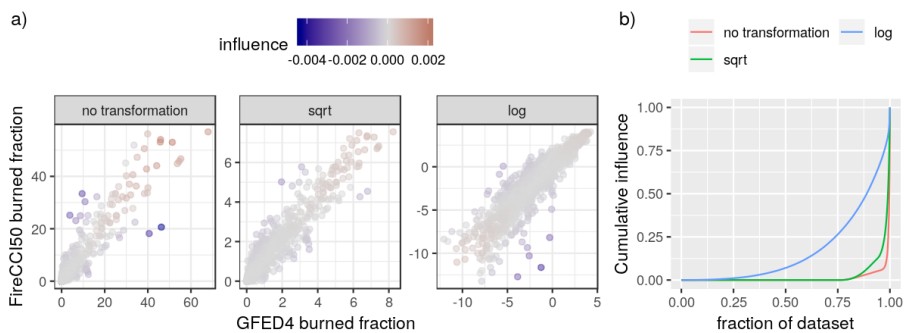

**Figure A9.** Scatter plots for the GFED4 and FireCCI50 dataset without transformation, square root transformation and log transformation (a), the color indicates the influence of individual data points on the correlation (computed as the difference in the correlation with and without that datapoint). Cumulative influence of data points in the dataset on the correlation (b). Without transformation a very small fraction has a strong influence on the correlation, these are grid cells with high burned area fraction (as can be seen in a).

**Table A1.** Reference literature for FireMIP models.

| Model | Land/ Vegetation model | Fire model |
|---|---|---|
| CLASS-CTEM | Arora and Boer (2005) Melton and Arora (2016) | Arora and Boer (2005) Melton and Arora (2016) |
| CLM | Oleson et al. (2013) | Li et al. (2012, 2013, 2014) |
| INFERNO | J. Best et al. (2011), Clark et al. (2011) | Mangeon et al. (2016) |
| JSBACH-SPITFIRE | Reick et al. (2013) | Lasslop et al. (2014) Hantson et al. (2015a) |
| LPJ-GUESS-SIMFIRE-BLAZE | Smith et al. (2001, 2014) Lindeskog et al. (2013) | Knorr et al. (2016) |
| LPJ-GUESS-SPITFIRE | Smith et al. (2001) Sitch et al. (2003) Ahlström et al. (2012) | Lehsten et al. (2009, 2015) |
| ORCHIDEE-SPITFIRE | Krinner et al. (2005) | Yue et al. (2014, 2015) |

**Table A2.** Correlation coefficients between burned area simulated by the FireMIP-models within the baseline experiment SF1 and the respective observation data. Due to the very skewed distribution of burned area, we use a square root transformation on both model and observations. Numbers in brackets show the Pearson correlation coefficients for not-transformed data. Only GFED4 and FireCCI50 provide uncertainty estimates, therefore GFED4s is not included. Correlation coefficients for 33% show the correlation between all grid points that lie within the 0–33% percentile of the relative standard error; values for 66% lie within the 33–66% percentile of the relative standard error and values for 99% lie within the 66–99% percentile. Bold numbers indicate correlation coefficients that are significant (p-value < 0.05).

| Model | GFED4 | | | FireCCI50 | | |
|---|---|---|---|---|---|---|
| | 33% | 66% | 99% | 33% | 66% | 99% |
| CLASS–CTEM | **0.59** (**0.41**) | **-0.08** (-0.07) | 0.04 (-0.03) | **0.58** (**0.38**) | -0.02 (-0.04) | 0.06 (0.003) |
| CLM | **0.78** (**0.72**) | **0.13** (**0.14**) | 0.09 (-0.03) | **0.80** (**0.73**) | **0.11** (**0.10**) | **0.09** (-0.03) |
| INFERNO | **0.76** (**0.68**) | **-0.18** (**-0.13**) | 0.05 (-0.02) | **0.77** (**0.64**) | -0.01 (0.01) | 0.05 (0.03) |
| JSBACH–SPITFIRE | **0.69** (**0.62**) | -0.08 (**-0.11**) | 0.02 (-0.05) | **0.68** (**0.56**) | -0.01 (-0.04) | 0.06 (0.01) |
| LPJ–GUESS–SIMFIRE–BLAZE | **0.70** (**0.55**) | -0.06 (-0.07) | -0.05 (**-0.10**) | **0.67** (**0.48**) | 0.03 (0.04) | -0.04 (-0.08) |
| LPJ–GUESS–SPITFIRE | **0.56** (**0.46**) | **0.42** (**0.41**) | **0.31** (**0.17**) | **0.61** (**0.48**) | **0.40** (**0.33**) | **0.47** (**0.34**) |
| ORCHIDEE–SPITFIRE | **0.82** (**0.74**) | **0.51** (**0.35**) | **0.48** (**0.36**) | **0.81** (**0.74**) | **0.49** (**0.31**) | **0.47** (**0.30**) |

*Author contributions.* LT and GL designed the study and performed the analysis with input from SPH, AH and SH. CY, GL, JM, LF, MF, SH provided simulations. LT, GL and SPH wrote the manuscript with contributions from all authors.

*Competing interests.* TEXT

*Disclaimer.* TEXT

5  *Acknowledgements.* The authors are grateful for the support and guidance of Silvia Kloster who initiated this work. We would like to thank Stephane Mangeon who performed the simulations with INFERNO. GL was funded by the Deutsche Forschungsgemeinschaft (DFG, German Research Foundation) – 338130981 and acknowledges the excellent computing support of DKRZ.

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
