# Peer review of "Response of simulated burned area to historical changes in environmental and anthropogenic factors: A comparison of seven fire models"

_Biogeosciences, 2019_

## Referee Comment (RC1) · Anonymous Referee #1 · 26 Mar 2019

The manuscript "Sensitivity of simulated historical burned area to environmental and anthropogenic controls: A comparison of seven fire models" by Teckentrup et al compares several global fire schemes implemented in different global land surface models in a controlled setup (based on FireMIP), to analyze which processes and parameterizations cause differences between models. To this end, the authors perform a sensitivity analysis, where five different factors ($CO_2$, population density, land use, lightning and climate) are individually modified. The authors identify land use as the most important factor for differences between models and discuss several potential routes to improve global fire models.

[Figure]

The manuscript represents a significant contribution to attempts to improve the parameterizations of Earth system models. It is well written and relatively easy to understand. I have, however, one major concern regarding the setup of the sensitivity analysis, which also effects a part of the findings presented in the manuscript (see comments below). This point should be accounted for before submitting a revised version.

General comments:

In my opinion, the design of the sensitivity analysis is not sufficient to support all conclusions made in the manuscript. The setup is suitable to analyze differences between models with respect to one factor (e.g. CO2). This is the case, because the modification of the factor (e.g. keep at constant value) is the same for all models, so differences between models have to result from the shape of the relation between this factor and the examined variable, burned area, which is implemented in the model. This is nicely explored in the manuscript by additional analyses of how the respective factors affect processes in the model.

However, the setup is not suitable to compare the relative effects, meaning the relative importance, of different factors, e.g. population dynamics and climate. The reason is that the factors show trends of different strength over the examined period (1900-2013). It is not clear to me how the authors separate the effect of the trend from the effect of the relation between factor and the simulated burned area (see specific comments below).

For example, let us assume that both CO2 and climate have a similar effect on burned area in the models. However, CO2 shows a strong trend in the period 1900-2013, while climate does not. This is enhanced in the setup of the sensitivity analysis by choosing a low value for CO2 for the experiment, but average values of climate variables. Consequently, the slope of the relative difference in burned area (e.g. Fig. 2) will be larger for CO2 than for climate, although both factors are (hypothetically) equally important in the model. This also affects the relative differences between models: If the general effect

[Figure]

of CO2 is amplified compared to climate in our hypothetical case, also the differences between models will be larger for CO2 than for climate. The authors need to clarify this, both in the methods and discussion section of the manuscript.

Specific comments:

P 2 L 7 Please replace 'regularly' by a more detailed description, such as 'at least once in 100 years' or similar. Does that mean that at least 60% of the land surface are never affected by fire?

P 2 L 12 Please put the 5.6 ppm CO2 into context: Which percentage of the total feedback per degree of warming does this correspond to?

P 2 L 26 Please explain the term 'woody thickening' shortly. How does vegetation composition change?

P 2 L 28 Why does reduced stomata conductance lead to increased fuel moisture? Is it assumed that plants take up water from the litter layer? Please explain this shortly.

P 3 L 6 It is quite difficult to understand this sentence. Please start with the end (nr of fires times size) and may be split into two sentences.

P 4 L 21 Does the around 150 year shorter spin-up for two of the models have effects on the fuel amount? Or is the turnover of the fuel fast enough to exclude that the models with shorter spin-up have less fuel?

P 5 Tab1 Why are only low values of CO2, population density and land use(?) included in the sensitivity analysis? Would it not make more sense to either use intermediate values, similar to climate and lightning, or, alternatively, test high values in addition to the low ones?

P 6 L 11 Please add a short description of how these data sets differ, beyond the retrieval algorithms, since this is important to understand the results (e.g. agricultural fires in GFED4s).

P 6 L 16 In which direction is the distribution skewed? Does the model resolution have an effect on the shape of the distribution?

P 6 L 21 The values 0.01 and 0.2 refer to the GFED4 and FireCCI50 data sets, I assume? Please make this clear.

P 8 L 9 - P9L2 I think this part should be shifted to the discussion.

P 9 L 4ff I do not understand the line of argument: In the first three experiments (CO2,population,land use), relatively strong trends and large model differences throughout the 20th century are reported. In the other two experiments, the trends are weaker. However, this result may be influenced from the setup of the sensitivity analysis, since there are trends in CO2, land use and population density over the 20th century. Population density, for instance, is kept at the low value of 1900 in the experiment, so it is logical that the rel. diff. BA increases over the 20th century for models, which assume a positive effect of population density on BA (e.g. LPJ-GUESS-SPITFIRE), due to the trend in population density. For models which assume a negative effect of population density on BA (e.g. LPJ-GUESS-SIMFIRE-BLAZE), the opposite is the case. However, it is not described how the effect of the trends (e.g. increase in population density) is separated from the effect of the factor in the model (e.g. effect of population density on fire). Figure 2 and Table 4 are only suitable to compare the relative effect of one factor between models, but not the relative importance of different factors. Maybe the relations between rel.diff. BA and lightning, and also rel.diff. BA and climate, are weak because the trends over the 20th century are not as pronounced as for the other factors, and also average values (1901-1920) are used for the experiments. In this case, the mean values of baseline scenario and the experiments would be very similar to each other, and variations would be randomly distributed over the 20th century, which is partly consistent with Fig. 2. Therefore, I am not convinced that the slope of the rel. diff. BA over the 20th century (Tab 4, Fig 2) is a good measure of the strength or importance of a certain factor in the model, compared to other factors.

P 12 L 11 Please add 'concentrations,' after 'CO2'.

P 16 L 3 Please explain shortly why the presence of lightning always leads to a net suppression of fire by humans.

P 18 L 15ff From the listed parameters, only the first two (precipitation and temperature) are climate variables. The others are dependent variables, which are also influenced by other factors (e.g. CO2). Please explain why you include them in the test. Moreover, I would like to see an analysis of the effects of wind speed. Is there a trend in wind speed from 1900 to 2013 ?

P 18 L 30 The word 'is' occurs one time too often.

P 19 L 10-12 I am not sure that this statement is valid, given my concerns on the setup of the sensitivity analysis above.

P 19 L 32 The word 'Table' is missing in the brackets.

P 21 L 14 How strong is the trend in changing climate compared to other trends, e.g. population density and CO2?

———————————————————

---

## Referee Comment (RC2) · Anonymous Referee #2 · 7 Apr 2019

General comments

The study is a useful compilation of the analysis of sensitivity experiments in the FireMIP output, but it is largely a technical report of the sensitivity of FireMIP model simulations of burned area since 1900. Philosophically, there is nothing really offered by the authors in terms of specific testing of improvements/changes needed with fire models beyond what has been pointed out in the literature in papers such as Van Marle et al 2017 and Andela et al 2017, and hinted at in the Hantson et al 2016 FireMIP overview paper and the Forkel et al 2019 paper.

While I appreciate the depth of the dissection of the causes for the discrepancies

among FireMIP models in this study, I find myself with no questions about FireMIP that have new or interesting answers, which is a concerning lack of momentum from the initially promising FireMIP effort. For example, did the FireMIP sensitivity experiments produce knowledge that the modeling groups could leverage for specific technical advances on, say, a future set of experiments? If anything, this paper makes me increasingly skeptical about the utility of FireMIP other than to show precisely what these authors stated in their conclusions: "Although burned area in most models compares reasonably well with satellite observations, there is a huge spread in transient simulations before the satellite era and a huge spread in the influence of the driving factors between models." Again, however, many FireMIP related papers have already pointed this out.

I recommend that the paper be published and I think that my comments fall somewhere between a minor and major revision, so I labeled it as minor revisions even though some of my comments might require some major discussion amongst the authors in terms of structuring a reply or rebuttal. The challenge that I offer to the authors is this: I do not see what we gain beyond now knowing that the sensitivity experiments are as confusingly inconclusive as the core experiments. If I were re-formulating my fire model and looking to this study, I would have little idea as to what the focus point should be other than simply acknowledging weaknesses such as the representation of human use of fire or needed better data for model parameterizations. The authors may need to make their case more clearly for this paper to stand out beyond being a technical report out.

Specific comments

Figures in the Supplement – please make larger versions of the maps in figures a1-a8. Another improvement would be to include a continuous rather than binary scale of values of the correlation coefficient in a2-a8. Painting the world with binary correlation coefficients would mask areas of potential weak and strong linear correlation. The strength of this study is the technical report-out of FireMIP sensitivity studies, so by

making figures a1-a8 so hard to read, the authors are undermining the very purpose of the work. Read another way, the community may gain more with more detail in the manuscript.

Page 6 line 16-17 – authors stated they used a square root transformation to reduce the skewness of the distribution, but it is unclear why. Please expand on both the reasons and what this transformation accomplishes. Perhaps a supplemental figure?

Page 6 line 19 – major uncertainties is a subjective phrasing that requires more qualifications. Humber et al 2018 clearly discussed the nuanced and important ways that observed burned area data sets agree and disagree when using global, regional, and varying temporal scales. Looking at Figure 3 in Humber et al 2018 and Figure 1 in this paper, however, the implication is that FireMIP models have even more than "major" uncertainties in the sense that even at an annual time scale, there is more spread amongst models than amongst the observations. Furthermore, the three burned area data sets discussed in this study (GFED4, GFED4s, and FireCCI50) show that there is agreement unless the specific methodological approach is augmented with the small fires approach described in Randerson et al 2012. Is that really a major disagreement or just a difference in analysis? Please be more specific or careful in the discussion around observational uncertainties. Also, please see my comment about Figure 1 below.

Page 6 line 20-21 – please explain what is meant by 0.01 and 0.2%. I am not following what the the values refer to.

Figure 1 would benefit from being split into a two-part plot: one part could remain as is, but the other would show the present day subset of the full analysis period. This is the evaluation period, but it is buried under too many curves.

Table 3 and page 7 – are these spatial correlation coefficients that compare the grid cell to grid cell agreement on a map? Or are they temporal correlation coefficients? It does not seem that Figure 1 temporal correlation is this high, but please clarify in the

text. If this is a spatial correlation, please include the figure in the Appendix as it could be valuable to modelers in identifying regional weaknesses in the FireMIP simulated burned area.

Table A2 is missing statistics relative to GFED4s.

Page 9 – the first sentence on this page highlights a major problem in the approach with modeling. Aiming at trends without a full understanding of the drivers in the simulations is .

Table 4 – while the M-K test is likely fine, the uncertainties (standard error or confidence intervals) in the slopes need to be included to understand the results better.

Page 9 and Section 3.2.4 – I thought that FireMIP only used a repeated lightning scaled to changes in modeled convection? While there is likely something to gain in the lightning sensitivity experiment, I would like to see some clearer discussion of the important caveats in interpreting the results. For example, would it be safe to surmise that there is no sensitivity to lightning changes since 1900 only if the modeled lightning is anything close to reality? Determining a lightning climatology from an untestable climate-model based parameterization and then drawing conclusions from that testing is prone to some circular or flawed logic.

Figure 2 – please re-title these with something that is easier to quickly interpret without cross-referencing the table. For example, I suggest (a) Constant CO2 (SF2_CO2), (b) Constant Population (SF2_FPO), (c) Constant Land Cover (SF2_FLA), (d) Constant Lightning (SF2_FLI), (e) Constant climate (SF2_CLI). Also please make figure 2 much wider to avoid the visual clutter of overlaid zigzagging lines.

Figure 2 – change the y-axes ranges so they are constant. It is hard to understand the sensitivity if the plotted range is variable.

Page 11 line 9 – I agree that the statistics suggest individual trends are significant but this does not preclude the massive spread (both positive and negative) in the trends

amongst models (table 4). I think this statement needs to include that caveat for an honest accounting of the FireMIP output.

Section 3.3 – the first paragraph makes no sense. What I am reading in this study is that the models barely agree on any trend, but yet the authors propose here that the models are important for understanding projected trends and supporting land management strategies. To me, a land management practice cannot be based on model trends that do not agree on trend and cannot be of much use if there is lack of agreement at country scales, let alone finer spatial scales.

Section 3.3, second paragraph – the results presented in the manuscript clearly show that models only agree in magnitude in the present day, but the quick microscope analysis of the present day trends show that observations and models do not agree in trends. Some models predict a positive slope, some negative. Unless the authors intend to propose that one FireMIP model is more physically realistic than another, then the results of the sensitivity studies are inconclusive.

Section 3.3 or 4 – it would be useful if these authors were to comment directly on fire models that did not contribute to FireMIP but that have contributed significantly to discussions of human-driven fire both in the present day and over the more distant past. This includes studies by Pfeiffer et al https://www.geosci-model-dev.net/6/643/2013/, Rabin et al https://www.geosci-model-dev.net/11/815/2018/, and Hantson et al https://journals.ametsoc.org/doi/full/10.1175/BAMS-D-15-00319.1 . All of these either echo or predict the results discussed by Andela et al 2017 and Bistinas et al 2014 related to a need to quantitatively represent the human use of fire on our planet in the modeling framework.

Conclusions – the conclusions are already evident in the Andela et al 2017 paper, so I do not see what we gain in this study. The authors conclude "further analyses are required to better disentangle" factors, but this is the same conclusion so many fire model and FireMIP papers have arrived at. Could the authors make a clearer argument

about what we gain in this manuscript?

---

## Author Comment (AC3) · 28 Jun 2019

In addition to the revisions addressing the reviewer comments, we found an inconsistency between the baseline and the SF2_CLI experiment with LPJ-GUESS-SPITFIRE. The source is at the moment still unknown. This sensitivity simulation was therefore removed in the revised manuscript.
* * *

---

## Author Response (AR1)

**Dear editor and referees,**

We want to thank you for your thoughts and comments on this manuscript. The reviews helped to clarify and improve the methodology, and reflect on the novel conclusions from this study compared to previous findings.

The major changes to the manuscript therefore are:

- A better explanation of the scope and novelty of this study (in the introduction, the discussion and conclusion sections)
- A clarification of the analysis of the trend in burned area and improved consistency between the different forcing factors

We below address the reviewer's comments point by point. We add *our replies in italic* and highlight suggested modifications in the manuscript in red. We number our replies and cross-refer to them to reduce the text if points had already been addressed before.

**Referee #1**

The manuscript "Sensitivity of simulated historical burned area to environmental and anthropogenic controls: A comparison of seven fire models" by Teckentrup et al compares several global fire schemes implemented in different global land surface models in a controlled setup (based on FireMIP), to analyze which processes and parameterizations cause differences between models. To this end, the authors perform a sensitivity analysis, where five different factors (CO2, population density, land use, lightning and climate) are individually modified. The authors identify land use as the most important factor for differences between models and discuss several potential routes to improve global fire models. The manuscript represents a significant contribution to attempts to improve the parameterizations of Earth system models. It is well written and relatively easy to understand. I have, however, one major concern regarding the setup of the sensitivity analysis, which also effects a part of the findings presented in the manuscript (see comments below). This point should be accounted for before submitting a revised version.

**General comments:**

In my opinion, the design of the sensitivity analysis is not sufficient to support all conclusions made in the manuscript. The setup is suitable to analyze differences between models with respect to one factor (e.g. CO2). This is the case, because the modification of the factor (e.g. keep at constant value) is the same for all models, so differences between models have to result from the shape of the relation between

this factor and the examined variable, burned area, which is implemented in the model. This is nicely explored in the manuscript by additional analyses of how the respective factors affect processes in the model. However, the setup is not suitable to compare the relative effects, meaning the relative importance, of different factors, e.g. population dynamics and climate. The reason is that the factors show trends of different strength over the examined period (1900-2013). It is not clear to me how the authors separate the effect of the trend from the effect of the relation between factor and the simulated burned area (see specific comments below). For example, let us assume that both CO2 and climate have a similar effect on burned area in the models. However, CO2 shows a strong trend in the period 1900-2013, while climate does not. This is enhanced in the setup of the sensitivity analysis by choosing a low value for CO2 for the experiment, but average values of climate variables. Consequently, the slope of the relative difference in burned area (e.g. Fig. 2) will be larger for CO2 than for climate, although both factors are (hypothetically) equally important in the model. This also affects the relative differences between models: If the general effect of CO2 is amplified compared to climate in our hypothetical case, also the differences between models will be larger for CO2 than for climate. The authors need to clarify this, both in the methods and discussion section of the manuscript.

- 1) We thank the reviewer for their assessment and the acknowledgement of our contributions. We address the methodological concerns by three points:
  - We agree with the reviewer that we do not separate the effect of the trend in the driver from the effect of the relation between factor and simulated burned area. We used the word term sensitivity loosely to mean the net response to the forcing, while the reviewer interprets it more formally as a change in response variable per unit change in forcing. To avoid confusion we adopt the reviewer's definition and thus have changed the title to "Response of simulated burned area to historical changes in environmental and anthropogenic factors: A comparison of seven fire models". As our goal was to understand which factors cause the response of burned area over the historical period we therefore need to look at the response given the present trends. Finding a high sensitivity for a forcing factor that has no trend would not directly help to understand the response over the historical period. We now reword the appropriate text passages accordingly and address which factors influenced the burned area over the historical period. Further, we highlight that response in burned area are caused by both: the sensitivity of the model and the imposed trend in the forcing. We also add the trends of the forcing datasets in the table 4 and include three sentences 'Response of simulated burned area to individual drivers' section:

The population density forcing dataset has the strongest trend in the relative differences between the transient forcing and the year 1920 value followed by the land-use and land cover change dataset. The trend in atmospheric CO2 concentration is higher than the trend in the lightning dataset, which is more than twice as strong as in the air temperature. Wind speed shows the lowest trend of all investigated driving factors (see tab. 4).

The reviewer notes that we use an average of the climate variables. This is not exactly what we did. We recycle the 20 first years that are available as climatic forcing (1900-1920) in the climate sensitivity simulations. However the reviewer is right that due to this there is no difference between the reference and the sensitivity simulation in the first 20 years of our comparison. We therefore now compute the trends of the in burned area between reference and sensitivity simulation starting in 1920 until the end of the simulation (2013). As we investigate the trend of differences with a consistent starting point for all factors (not simply the differences between sensitivity and reference simulation) we can now also compare the importance between the factors for the simulated historical changes of burned area.

We add in the manuscript in the Methods:

these changes.

The resulting difference in burned area between the simulations is then a combination of the changes in the forcing and the sensitivity of the model to that forcing factor.

and in the Response of simulated burned area to individual drivers section (see also reply 21): The response of burned area to the individual factors is determined by the changes in the driving factors and the sensitivity of the model to

We use the word sensitivity now only in these places and for "sensitivity experiment". In other places sensitivity has been replaced with "response of simulated burned area to".

• As a second change we now use the absolute differences instead of relative differences. As the CO2 concentration for instance was fixed at the value of 1750, for some models the burned area that is used to normalized is much smaller than it would be if the value was set to the value of 1900. All models have a comparable magnitude of burned area for present day therefore the absolute changes are also comparable and the comparison between models is not strongly influenced. The reviewer did not directly request this but we think that this increases the comparability between the factors. Our conclusions are not affected by this change but the quantification of trends is more meaningful. We add in the Methods section

Two of the models (CLASS--CTEM and CLM) started the simulations later than the others (1861 and 1850, respectively) and due to limitations in data availability the reference year of the forcings used in the spin-up varies (see tab. 1). We account for these differences in starting years between models and of the forcing factors by limiting our analysis to the period where all factors are different from the ones used in the spin-up (after 1921). These differences still influence the absolute differences, we therefore quantify the strength of the impact through the slope of a regression line and do not interpret the offset.

**Specific comments:**

P 2 L 7 Please replace 'regularly' by a more detailed description, such as 'at least once in 100 years' or similar. Does that mean that at least 60% of the land surface are never affected by fire?

2) The descriptions in the literature were not hat precise, thus we have removed the sentence.

P 2 L 12 Please put the 5.6 ppm CO2 into context: Which percentage of the total feedback per degree of warming does this correspond to?

3) We now include the strength of the global land climate-carbon-cycle feedback (17.5 ppm K-1) as a context. It corresponds to a percentage of approximately 32%.

Analyses based on observations of the pre-industrial period suggest that the contribution of fire to the overall climate–carbon-cycle feedback is substantial with 5.6 ± 3.2 ppm K-1 CO2 (Harrison et al., 2018) while the strength of the global land climate–carbon-cycle feedback estimated from Earth system simulations (Arora et al., 2013) is 17.5 ppm K-1 (Harrison et al., 2018). However, comparing potential fire-induced losses from terrestrial carbon pools and stocks of solid pyrogenic carbon in soils and ocean, fire may also be a net sink of carbon and Earth system simulations show a negative effect of fire on radiative forcing (Lasslop et al., 2019).

P 2 L 26 Please explain the term 'woody thickening' shortly. How does vegetation composition change?

4) We modified the manuscript as follows:

It can lead to an increase in the abundance of woody plants ('woody thickening'; Wigley et al., 2010; Bond and Midgley, 2012; Buitenwerf et al., 2012) [...]

P 2 L 28 Why does reduced stomata conductance lead to increased fuel moisture? Is it assumed that plants take up water from the litter layer? Please explain this shortly.

5) It is assumed that the water saving increases soil moisture and in consequence fuel moisture, including the living biomass contribution to the fuel load and the amount of litter on the soil surface.

On the other hand, decreased stomatal conductance and lower transpiration can lead to enhanced water conservation in plants. This increases the moisture content of soil as well as vegetation moisture content and consequently live and dead fuel moisture contents, which decreases flammability and in consequence reduces burned area.

P 3 L 6 It is quite difficult to understand this sentence. Please start with the end (nr offires times size) and may be split into two sentences.

6) We rephrased the sentence:

Burned area can be expressed as the number of fires multiplied by their fire size. The increase in burned area due to changes in ignitions is expected to differ between regions with varying population density as the largest fires occur in unpopulated areas (Hantson et al., 2015a).

P 4 L 21 Does the around 150 year shorter spin-up for two of the models have effects on the fuel amount? Or is the turnover of the fuel fast enough to exclude that the models with shorter spin-up have less fuel?

7) The described simulations start from a spinup simulation where carbon pools were equilibrated. We add a sentence to describe this point in the Methods section:

The baseline FireMIP experiment (SF1) is a transient simulation from 1700-2013, in which atmospheric CO2 concentration, population density, land-use, lightning, and climate change through time according to prescribed datasets. The baseline and sensitivity simulations start from the end of a spin-up simulation with equilibrated carbon pools (see Rabin et al. (2017a) for details of the experimental protocol).

P 5 Tab1 Why are only low values of CO2, population density and land use(?) included in the sensitivity analysis? Would it not make more sense to either use intermediate values, similar to climate and lightning, or, alternatively, test high values in addition to the low ones?

8) See also reply 1. The experiments were designed to understand the influence of the historical variation in the driving factors on the simulated burned area. Therefore all factors were individually held constant at the initial conditions, e.g. the conditions that were used in the spin-up. Lightning and

climate varied in the historical baseline simulation from 1900 and were set to the first twenty years before, as no forcing dataset is available before that time and because the interannual variability in climate is important (so using only one year is not an option). We now compute the trends starting with the year 1920, when all factors vary. Results may be slightly different when fixing the forcing at values of different years, but as we are interested in how the historical changes influenced the historical simulations in burned area we think the interpretation of the high values would be less direct. The sensitivity simulations now start with a state that existed in the past (neglecting, of course, any existing errors in the models and forcing datasets). Starting the simulation with the high values would be a hypothetical case, as the models also slightly depend on their history. Technically this would also mean that the sensitivity simulations all require a separate spin-up. They would start from different initial conditions and although they would end with the same forcing the model state would likely be different as for present day ecosystems are not in equilibrium due to global change.

P 6 L 11 Please add a short description of how these data sets differ, beyond the retrieval algorithms, since this is important to understand the results (e.g. agricultural fires in GFED4s)

9) We now include an improved description how these datasets differ. To evaluate the simulations of burned area, we compare the simulated burned area with remote sensing data products. Global burned area observations from satellites still suffer from substantial uncertainty, as reflected by the considerable differences in spatial and temporal patterns between different data products (Humber et al., 2018; Hantson et al., 2016a; Chuvieco et al., 2018; van der Werf et al., 2017). Using multiple satellite products in model benchmarking is one approach to take into account these observational uncertainties (Rabin et al., 2017a). In this study, we use three satellite products: GFED4 (Giglio et al., 2013), GFED4s (van der Werf et al., 2017) and FireCCI50 (Chuvieco et al., 2018). GFED4 is a gridded version of the MODIS Collection 5.1 MCD64 burned area product. It is known that this product strongly underestimates small fires, including cropland fires (e.g.Hall et al. (2016)). In GFED4s, burned area due to small fires is estimated based on MODIS active fire (AF) detections and added to GFED4 burned area. However, this methodology may introduce significant errors related to erroneous AF detections (Zhang et al., 2018). As a complementary product, FireCCI50 was developed using MODIS spectral bands with higher spatial resolution than MCD64. A higher resolution enhances the ability to detect smaller fires; however, this improvement is partially offset by suboptimal spectral properties of the bands. Both GFED4s and FireCCI50 have larger burned area than GFED4. Since all three products are based on MODIS data, the inter-product

differences probably underestimate uncertainties associated with these products. A recent mapping of burned area for Africa using higher resolution Sentinel-2 observations indicates that all three products substantially underestimate burned area (Roteta et al., 2019). For the model evaluation we use temporally averaged burned area fraction for the years 2001–2013, the interval common to all three satellite products and the model simulations.

Hall, J. V., T. V. Loboda, L. Giglio and G. W. McCarty (2016). "A MODIS-based burned area assessment for Russian croplands: Mapping requirements and challenges." Remote sensing of environment 184: 506-521.

Roteta, E., A. Bastarrika, M. Padilla, T. Storm and E. Chuvieco (2019). "Development of a Sentinel-2 burned area algorithm: Generation of a small fire database for sub-Saharan Africa." Remote Sensing of Environment 222: 1-17.

Zhang, T., Wooster, M., de Jong, M., and Xu, W.: How Well Does the 'Small Fire Boost' Methodology Used within the GFED4.1s Fire Emissions Database Represent the Timing, Location and Magnitude of Agricultural Burning?, Remote Sensing, 10, 823, https://doi.org/10.3390/rs10060823, 2018.

P 6 L 16 In which direction is the distribution skewed? Does the model resolution have an effect on the shape of the distribution?

10) The distribution of burned area has a very large fraction of 0 and small burned area, high fractions of burned area have a very low frequency. We add a plot indicating the influence of individual datapoint in the comparison between GFED4 and FireCCI50 in the supplement. Without transformation a very small fraction of the data points determines the correlation, this is improved with the squareroot transformation and would be further improved using a log transformation, but that would mean that grid cells with 0 would be excluded. As the correlation should provide a global evaluation of the model a much higher influence of individual grid cells is not desirable. As the models are all aggregated to the same spatial resolution the model resolution does not have an influence on the distribution.

Figure A9: Scatter plots for the GFED4 and FireCCI50 dataset without transformation, square root transformation and log transformation (a), the color

indicates the influence of individual data points on the correlation (computed as the difference in the correlation with and without that datapoint). Cumulative influence of data points in the dataset on the correlation (b). Without transformation a very small fraction has a strong influence on the correlation, these are grid cells with high burned area fraction (as can be seen in a).

**We also modify the text in the main paper:**

We quantify the agreement between models and observations by providing the global burned area and the Pearson correlation coefficient for the between grid cell variation (see tab. 3). We choose the Pearson correlation as it quantifies the covariation of the spatial patterns, and is less sensitive to the highly uncertain absolute burned area values. Burned area has a strongly skewed distribution, with few high values and many small values close to, or equal to, zero. These few high values have a much higher contribution to the overall correlation (see figure A9 in Appendix) and therefore the metric is strongly determined by the performance of the model in areas with high burning. Square root or logarithmic transformation leads to more normally distributed values, that reduce this bias (see figure A9 in Appendix). As the logarithm transformation excludes grid cells with zero burned area, we adopt the square root transformation.

P 6 L 21 The values 0.01 and 0.2 refer to the GFED4 and FireCCI50 data sets, I assume? Please make this clear.

11) We clarify in the manuscript [...] yields uncertainty estimates of 0.01% (GFED4) and 0.2% (Fire CCI50)

P 8 L 9 - P 9 L 2 I think this part should be shifted to the discussion.

12) We did not separate Results and Discussion but directly discuss the results following the presentation. We shortened the indicated paragraphs slightly to have more emphasis on the results and moved part of it to the "Implications for model development and applications" section.

P 9 L 4ff I do not understand the line of argument: In the first three experiments (CO2,population,land use), relatively strong trends and large model differences throughout the 20th century are reported. In the other two experiments, the trends are weaker. However, this result may be influenced from the setup of the sensitivity analysis, since there are trends in CO2, land use and population density over the 20th century. Population density, for instance, is kept at the low value of 1900 in the experiment, so it is logical that the rel. diff. BA increases over the 20th century for models, which assume a positive effect of population density on BA (e.g. LPJ-GUESS-SPITFIRE), due to the trend in population density. For models which

assume a negative effect of population density on BA (e.g.

LPJ-GUESS-SIMFIRE-BLAZE), the opposite is the case.

However, it is not described how the effect of the trends (e.g. increase in population density) is separated from the effect of the factor in the model (e.g. effect of population density on fire).

13) See also reply 1). Population density is kept at the value of 1700. We now use the absolute differences. The initial values of land use, CO2 and climate stem from different years. This is because climate data were only available from 1900 onwards. We now compute the trends starting in 1920 when all factors vary, with low influence on the results. Fig. 2 already showed the strong interannual variability of climate and lightning and the absence of trends over the whole period. Qualitatively the spread between models for population density is logical considering the different assumptions in the models, but note that most models assume a curve with a maximum and therefore include positive and negative effects. Quantification of the net effect and also the magnitude of the effect therefore requires the sensitivity simulations provided in this study. As we aim to quantify the effect of forcing factors over the simulation period we quantify the response in burned area given the historical trend. Quantification of the burned area response with a hypothetical trend (for instance a doubling) would not allow to understand the historical simulated trends.

Figure 2 and Table 4 are only suitable to compare the relative effect of one factor between models, but not the relative importance of different factors. Maybe the relations between rel.diff. BA and lightning, and also rel.diff. BA and climate, are weak because the trends over the 20th century are not as pronounced as for the other factors, and also average values (1901-1920) are used for the experiments. In this case, the mean values of baseline scenario and the experiments would be very similar to each other, and variations would be randomly distributed over the 20th century, which is partly consistent with Fig. 2. Therefore, I am not convinced that the slope of the rel. diff. BA over the 20th century (Tab 4, Fig 2) is a good measure of the strength or importance of a certain factor in the model, compared to other factors.

14) We now use the absolute differences, see reply 1. We assume this may also again relate to the fact that we did not separate out the strength of the trend in the driving factor. See previous comment and reply 1 and 8. We now clarify that we are interested to understand which factors cause the simulated trends over the historical period. Note that the climate was not averaged over the 1900-1920 period but recycled. We now compute the trends for the absolute differences and for the period 1920 to 2013 for which all factors vary.

P 12 L 11 Please add 'concentrations,' after 'CO2'.

15) We replaced all occurrences of 'CO2' with 'atmospheric CO2 concentration' to be precise.

P 16 L 3 Please explain shortly why the presence of lightning always leads to a net suppression of fire by humans.

16) The effect of increasing human ignitions is strongest if no other ignitions are present. If lightning already ignited a fire and additional human ignition has little effect. This was tested with the CTEM model, which is also part of this intercomparison study. We include in the text:

The presence of lightning ignitions reduces the limiting effect of a lack of human ignitions on burned area. For the CLASS-CTEM model as soon as lightning ignitions are present, the net effect of humans is to suppress fires, even though the underlying relationship assumes an increase in ignitions with population density (Arora and Melton, 2018, supplement). This may explain why global models assuming an increase of ignitions with increases in population density are able to capture the burned area variation along population density gradients (Lasslop and Kloster, 2017; Arora and Melton, 2018) and why global statistical analyses find a net human suppression also for low population density (Bistinas et al., 2014).

P 18 L 15ff From the listed parameters, only the first two (precipitation and temperature) are climate variables. The others are dependent variables, which are also influenced by other factors (e.g. CO2). Please explain why you include them in the test. Moreover,I would like to see an analysis of the effects of wind speed. Is there a trend in wind speed from 1900 to 2013 ?

17) We include the vegetation parameters in addition to the climate parameters as climate influences fire not only directly but also through its influence on vegetation. We modify the included explanation: "The influence of climate on burned area is complex; it influences burned area through the meteorological conditions and through effects on vegetation conditions that influence fuel load and fuel characteristics (Scott et al., 2014). We therefore correlated for each grid cell changes in physical parameters (precipitation, temperature, wind speed and soil moisture) and vegetation parameters (litter, vegetation carbon and grass biomass) with changes in burned area." Note that CO2 is not different between the simulations compared here, only climate differs. In addition, we add the linear regression slope and the standard deviation for wind speed in table 4; over 1921 - 2013, the relative difference in wind speed has a significant negative linear regression slope (-0.012 +- 0.006). We add 'Wind speed shows the lowest trend of all investigated driving factors (see tab. 4).'

P 18 L 30 The word 'is' occurs one time too often. *18) Removed.*  P 19 L 10-12 I am not sure that this statement is valid, given my concerns on the setup of the sensitivity analysis above.

19) See reply 1, 8, 13, 14. This refers to "Representing human influence on fire is the major challenge for long-term projections. Our analyses of the controls on the variability of fire suggest that human activities drive the long term (decadal to centennial) trajectories, while considering climate variability may be sufficient for short-term projections."

We have now improved the computation of trends. To assess the importance of certain factors in trajectories the underlying trend is important, a separation of the trend in forcing from the sensitivity of the model would therefore not improve the assessment. However changes in the trends of the forcing factors for future can change the results we therefore included:

Changes in the trends of the driving factors may change this balance. For instance, stronger changes in climate into the future may increase the relative importance of climate for long term fire projections in the future.

P 19 L 32 The word 'Table' is missing in the brackets. *20) It is included now.*

P 21 L 14 How strong is the trend in changing climate compared to other trends, e.g.population density and CO2?

21) We now quantify the trends in the forcing factors. It is however questionable how comparable these changes are between factors. Also the global increases in CO2 are more meaningful than global changes in temperature as CO2 is fairly similar in different locations while the changes in temperature vary regionally. For text modifications, see reply 1.

**Dear editor and referees,**

We want to thank you for your thoughts and comments on this manuscript. The reviews helped to clarify and improve the methodology, and reflect on the novel conclusions from this study compared to previous findings.

The major changes to the manuscript therefore are:

- A better explanation of the scope and novelty of this study (in the introduction, the discussion and conclusion sections)
- A clarification of the analysis of the trend in burned area and improved consistency between the different forcing factors

We below address the reviewer's comments point by point. We add *our replies in italic* and highlight suggested modifications in the manuscript in red. We number our replies and cross-refer to them to reduce the text if points had already been addressed before.

**Referee #2**

**General comments**

The study is a useful compilation of the analysis of sensitivity experiments in the FireMIP output, but it is largely a technical report of the sensitivity of FireMIP model simulations of burned area since 1900. Philosophically, there is nothing really offered by the authors in terms of specific testing of improvements/changes needed with firemodels beyond what has been pointed out in the literature in papers such as Van Marle et al 2017 and Andela et al 2017, and hinted at in the Hantson et al 2016 FireMIP overview paper and the Forkel et al 2019 paper. While I appreciate the depth of the dissection of the causes for the discrepancies among FireMIP models in this study, I find myself with no questions about FireMIP that have new or interesting answers, which is a concerning lack of momentum from the initially promising FireMIP effort. For example, did the FireMIP sensitivity experiments produce knowledge that the modeling groups could leverage for specific technical advances on, say, a future set of experiments? If anything, this paper makes me increasingly skeptical about the utility of FireMIP other than to show precisely what these authors stated in their conclusions: "Although burned area in most models compares reasonably well with satellite observations, there is a huge spread in transient simulations before the satellite era and a huge spread in the influence of the driving factors between models." Again, however, many FireMIP related papers have already pointed this out. I recommend that the paper be published and I think that my comments fall somewhere between a minor and major revision, so I labeled it as

minor revisions even though some of my comments might require some major discussion amongst the authors in terms of structuring a reply or rebuttal. The challenge that I offer to the authors is this: I do not see what we gain beyond now knowing that the sensitivity experiments areas confusingly inconclusive as the core experiments. If I were re-formulating my firemodel and looking to this study, I would have little idea as to what the focus point should be other than simply acknowledging weaknesses such as the representation of human use of fire or needed better data for model parameterizations. The authors may need to make their case more clearly for this paper to stand out beyond being a technical report out.

1) We thank the reviewer for the critical review and take the chance to reflect and rework our conclusions. We include improvements in the Introduction, the discussion and the conclusions to clarify the novelty of our study. In the introduction we clarify how our work relates to previous work: Fire-enabled vegetation models simulate fire regimes in response to the combination of individual forcings, including atmospheric CO2 concentration, population density, land-use change, lightning and climate. Individual fire-enabled vegetation models have been shown to simulate observed global patterns of burned area and fire emissions reasonably well (Kloster et al., 2010; Prentice et al., 2011; Li et al., 2012; Lasslop et al., 2014; Yue et al., 2014), but there are large differences between models in terms of regional patterns, fire seasonality and interannual variability, historical trends (Kelley et al., 2013; Andela et al., 2017) and responses to individual factors (Kloster et al., 2010; Knorr et al., 2014, 2016; Lasslop and Kloster, 2017, 2015). The fire model intercomparison project (FireMIP, Hantson et al., 2016a; Rabin et al., 2017a) provides a systematic framework to consistently analyse and understand the causes of these differences and to relate them to differences in the treatment of key drivers of fire in individual models. The FireMIP project provides simulations for a systematic comparison of fire-model behaviour based on outputs of a large range of models with identical forcing inputs. In addition to a reference historical simulation, sensitivity simulations were conducted for individual forcings, specifically atmospheric CO2 concentration, population density, land-use change, lightning and climate. A recent evaluation of the FireMIP models indicates that the relationship with climatic parameters is captured well by models, the response to human factors is captured by some models and the response to vegetation productivity or the allocation of carbon to fuels needs refinement for most models (Forkel et al., 2019a). Comparisons of the FireMIP historical simulations found differences in transient model behaviour in the 20th century (Andela et al., 2017; van Marle et al., 2017). The causes of the differences and the reasons why different models show different responses are not yet understood.

Our study shows in detail which model responses of burned area to environmental factors can be understood, how these are related to the model equations and how these translate into certain trends of burned area. The understanding on how certain model assumptions lead to trends in burned area is novel, the need for this was emphasized by the previous publications (but they do not provide it) and the recently detected trends in the satellite data. We improved the sections discussing the new possibilities for model reparameterization:

[revised manuscript text omitted]

Followed by the summary of insights for the individual factors. We add for the effect of population density:

It would be useful to develop an approach that represents local human-fire relationships, but this will likely remain a long term challenge and requires the synthesis of knowledge from various research fields.

We add for the effect of land use and land cover change:

Improved knowledge on the effects of land-use intensity on burned area and the development of appropriate forcing datasets could strongly support model development.

**And end with:**

The uncertainties in global fire models need to be taken into account in model applications, for instance if model simulations are to be used to support climate adaptation strategies. Model ensemble simulations can give indications of such uncertainties. Therefore the results of this study provide a basis to interpret uncertainties in global fire modelling studies. The spatial patterns of burned area and its drivers are already well explored and understood. We here provide a summary of which model assumptions need additional constraints to efficiently reduce the uncertainty in temporal trends.

**Specific comments**

Figures in the Supplement – please make larger versions of the maps in figures a1-a8. Another improvement would be to include a continuous rather than binary scale of values of the correlation coefficient in a2-a8. Painting the world with binary correlation coefficients would mask areas of potential weak and strong linear correlation. The strength of this study is the technical report-out of FireMIP sensitivity studies, so by making figures a1-a8 so hard to read, the authors are undermining the very purpose of the work. Read another way, the community may gain more with more detail in the manuscript.

2) Figure a2-a8 are not correlations but the slope coefficients. It only shows significant changes to identify regions with weak relationships. We wanted to emphasize the spatial distribution of decreases and increases and therefore chose this color scale. We now provide the graphs with the more detailed color scale and larger versions of the maps, because, as the reviewer suggests, it will be useful for the community.

Page 6 line 16-17 – authors stated they used a square root transformation to reduce the skewness of the distribution, but it is unclear why. Please expand on both the reasons and what this transformation accomplishes. Perhaps a supplemental figure?

3) See also reply 10 for reviewer 1. The correlation coefficient is most useful for normally distributed variables. The burned area varies over several orders of magnitude and the skewed distribution gives the highest importance to values with very high burned area. We transformed the data to improve the applicability of the metric. We include now a figure illustrating the influence of individual data points to the correlation, showing that the outliers in the untransformed data have a really high contribution and determine the correlation (figure A9 in the Appendix). This is improved with the squareroot transformation and would be further improved using a log transformation, but that would mean that grid cells with 0 would be excluded. With the transformation the contribution is better distributed to all data points, it is therefore more useful for global modelling where a too strong focus on only grid cells with high burned area can be distracting.

Figure A9: Scatter plots for the GFED4 and FireCCI50 dataset without transformation, square root transformation and log transformation (a), the color indicates the influence of individual data points on the correlation (computed as the difference in the correlation with and without that datapoint). Cumulative influence of data points in the dataset on the correlation (b). Without transformation a very small fraction has a strong influence on the correlation, these are grid cells with high burned area fraction (as can be seen in a).

We also modify the text in the main paper:

We quantify the agreement between models and observations by providing the global burned area and the Pearson correlation coefficient for the between grid cell variation (see tab. 3). We choose the Pearson correlation as it quantifies the covariation of the spatial patterns, and is less sensitive to the highly uncertain absolute burned area values. Burned area has a strongly skewed distribution, with few high values and many small values close to, or equal to, zero. These few high values have a much higher contribution to the overall correlation (see figure A9 in Appendix) and therefore the metric is strongly determined by the performance of the model in areas with high burning. Square root or logarithmic transformation leads to more normally distributed values, that reduce this bias (see figure A9 in Appendix). As the logarithm transformation excludes grid cells with zero burned area, we adopt the square root transformation.

Page 6 line 19 – major uncertainties is a subjective phrasing that requires more qualifications. Humber et al 2018 clearly discussed the nuanced and important ways that observed burned area data sets agree and disagree when using global, regional, and varying temporal scales. Looking at Figure 3 in Humber et al 2018 and Figure 1 in this paper, however, the implication is that FireMIP models have even more than "major" uncertainties in the sense that even at an annual time scale, there is more spread amongst models than amongst the observations. Furthermore, the three burned area data sets discussed in this study (GFED4, GFED4s, and FireCCI50) show that there is agreement unless the specific methodological approach is augmented with the small fires approach described in Randerson et al 2012. Is that really a major disagreement or just a difference in analysis? Please be more specific or careful in the discussion around observational uncertainties. Also, please see my comment about Figure 1 below.

4) See also reply 9 for reviewer 1. In Figure 1, the models are largely within the range of the observations for the evaluation period. The section shows that the models are largely in the range of satellite observed burned area and have a reasonable spatial distribution (see appendix figure A1). There is methodological uncertainty in satellite burned area products and this is reflected in the variation between the products due to the methodological approach applied. The spread between these products still underestimates the uncertainty in the satellite products as all are based on the same sensor (MODIS). This is already mentioned in the manuscript on p.6 I. 23. We improve the paragraph with more details on the differences between the sensors and also link it to more recent burned area estimation using the high resolution Sentinel-2 data, which gives insights in the huge uncertainty of satellite products (see also reply 9 for reviewer 1).

To evaluate the simulations of burned area, we compare the simulated burned area with remote sensing data products. Global burned area observations from satellites still suffer from substantial uncertainty, as reflected by the considerable differences in spatial and temporal patterns between different data products (Humber et al., 2018; Hantson et al., 2016a; Chuvieco et al., 2018; van der Werf et al., 2017). Using multiple satellite products in model benchmarking is one approach to take into account these observational uncertainties (Rabin et al., 2017a). In this study, we use three satellite products: GFED4 (Giglio et al., 2013), GFED4s (van der Werf et al., 2017) and FireCCI50 (Chuvieco et al., 2018). GFED4 is a gridded version of the MODIS Collection 5.1 MCD64 burned area product. It is known that this product strongly underestimates small fires, including cropland fires (e.g.Hall et al. (2016)). In GFED4s, burned area due to small fires is estimated based on MODIS active fire (AF) detections and added to GFED4 burned area. However, this methodology may introduce significant errors related to erroneous AF detections (Zhang et al., 2018). As a complementary product, FireCCI50 was developed using MODIS spectral bands with higher spatial resolution than MCD64. A higher resolution enhances the ability to detect smaller fires; however, this improvement is partially offset by suboptimal spectral properties of the bands. Both GFED4s and FireCCI50 have larger burned area than GFED4. Since all three products are based on MODIS data, the inter-product differences probably underestimate uncertainties associated with these products. A recent mapping of burned area for Africa using higher resolution Sentinel-2 observations indicates that all three products substantially underestimate burned area (Roteta et al., 2019). For the model evaluation we use temporally averaged burned area fraction for the years 2001–2013, the interval common to all three satellite products and the model simulations.

Hall, J. V., T. V. Loboda, L. Giglio and G. W. McCarty (2016). "A MODIS-based burned area assessment for Russian croplands: Mapping requirements and challenges." Remote sensing of environment 184: 506-521.

Roteta, E., A. Bastarrika, M. Padilla, T. Storm and E. Chuvieco (2019). "Development of a Sentinel-2 burned area algorithm: Generation of a small fire database for sub-Saharan Africa." Remote Sensing of Environment 222: 1-17.

Zhang, T., Wooster, M., de Jong, M., and Xu, W.: How Well Does the 'Small Fire Boost' Methodology Used within the GFED4.1s Fire Emissions Database Represent the Timing, Location and Magnitude of Agricultural Burning?, Remote Sensing, 10, 823, https://doi.org/10.3390/rs10060823, 2018.

Moreover we now include a new publication (Forkel et al. 2019) in the discussion which shows that the trends as observed by satellites are still highly uncertain and not robust.

Satellite records show a decline in global burned area since 1996 (Andela et al., 2016). However, as Forkel et al. (2019b) have shown, the significance of the observed global decline is strongly affected by the length of the sampled interval because of the high interannual variability in burned area and trends between products show only a low correlation (Forkel et al., 2019b). No observations document the longer term trends in burned area. Charcoal records (Marlon et al., 2008, 2016) and carbon monoxide data from ice-core records (Wang et al., 2010) are a proxy for biomass burning and show a global decrease in biomass burning over most of the 20th century. However, the charcoal records show an increase in burning since 2000 CE, but this discrepancy might reflect regional undersampling (for instance in Africa) or taphonomic issues of the charcoal record. A recent fire emission dataset (van Marle et al., 2017) merges information from satellites, charcoal records, airport visibility records and if no other information was available uses simulation results of the FireMIP models. This dataset is not included to evaluate the models here as it is partly based on the simulations of the FireMIP models and as it provides only estimates for emissions not burned area. The understanding of the drivers on simulated trends that we give below provides insights on what causes the simulated trends and which assumptions control the trend. These insights will help to understand which observational constraints and process understanding is required to improve global fire models.

Page 6 line 20-21 – please explain what is meant by 0.01 and 0.2%. I am not following what the values refer to.

5) We clarify in the manuscript, see also reply 11 for reviewer 1: [...] yields uncertainty estimates of 0.01 % (GFED4) and 0.2% (Fire CCI50)

Figure 1 would benefit from being split into a two-part plot: one part could remain asis, but the other would show the present day subset of the full analysis period. This is the evaluation period, but it is buried under too many curves.

6) Unfortunately this suggestion would lead to us exactly reproducing the figure number 3 of the Andela et al 2017 paper and contradicts the general suggestion of the reviewer to go beyond previous studies. We do agree, however, that the satellite datasets are buried under the curves in our plot. We now include a shaded area for the range of the satellite datasets as this is the main point we wish to convey here. As well, since we do not want to focus on evaluation of the models (which has been the focus of Andela et al. 2017 and Forkel et al. 2019 already) we rephrase the heading of this section to "Simulated historical burned area" to reflect the focus on the longer term trends and understanding the reasons for the divergence between models, independent of their correctness. We add a reference to Forkel et al. (2019) for more details.

Table 3 and page 7 – are these spatial correlation coefficients that compare the gridcell to grid cell agreement on a map? Or are they temporal correlation coefficients? It does not seem that Figure 1 temporal correlation is this high, but please clarify in the text. If this is a spatial correlation, please include the figure in the Appendix as it could be valuable to modelers in identifying regional weaknesses in the FireMIP simulated burned area.

7) We conduct a gridcell to gridcell comparison here, however spatial correlation coefficient is not a statistical term and may be confused with spatial auto-correlation. It implies some consideration of the geographical location. For table 3, we average burned area fraction over 2001 - 2013 (compare figure A1) and then correlate all individual grid cells of the remotely sensed product with the respective model. Therefore there is only one value, we did not analyse the spatial distribution or regional variation. For example, the first value in table 3, column 'R(GFED4, model)' is the Pearson correlation coefficient between the square root-transformed burned area fraction averaged over 2001 - 2013 in GFED4 and the square root-transformed burned area fraction averaged over 2001 - 2013 in CLASS-CTEM. We now include the "correlation over grid cells" to indicate it is not over time and change the caption of table 3 to "Global burned area averaged over 2001–2013 in Mha yr-1 and the Pearson correlation coefficients between burned area fraction averaged over 2001 - 2013 in the baseline experiment SF1 for all FireMIP-models and the respective observation data over all grid cells. We use a square root transformation on both model and observations. All correlation coefficients are significant (p-value < 0.05).

Table A2 is missing statistics relative to GFED4s.

8) GFED4s does not provide uncertainty estimates and therefore is not included in table A2. (We change the table caption from 'GFED4 and FireCCI50 provide uncertainty estimates' to 'Only GFED4 and FireCCI50

provide uncertainty estimates, therefore GFED4s is not included' to clarify this.)

Page 9 – the first sentence on this page highlights a major problem in the approach with modeling. Aiming at trends without a full understanding of the drivers in the simulations is .

9) One sentence in this comment is incomplete. It refers to the following sentence "The better understanding of the drivers of simulated trends that we provide below can inform us on how certain trends can be achieved in models." We speculate that the reviewer wants to indicate, that the possibility to achieve a trend based on a certain driver, does not necessarily mean that this is correct. Being aware however of how trends can be achieved is a useful information for model development. Whether the changes are plausible still needs to be addressed before implementing them. *We add*:

The understanding of the drivers on simulated trends that we give below provides insights on what causes the simulated trends and which assumptions control the trend. These insights will help to understand which observational constraints and process understanding is required to improve global fire models.

Table 4 – while the M-K test is likely fine, the uncertainties (standard error or confidence intervals) in the slopes need to be included to understand the results better.

10) We include the uncertainties of the slope parameter. However the Mann-Kendall test is better suited to understand whether the trend is significant.

Page 9 and Section 3.2.4 – I thought that FireMIP only used a repeated lightning scaled to changes in modeled convection? While there is likely something to gain in the lightning sensitivity experiment, I would like to see some clearer discussion of the important caveats in interpreting the results. For example, would it be safe to surmise that there is no sensitivity to lightning changes since 1900 only if the modeled lightning is anything close to reality? Determining a lightning climatology from an untestable climate-model based parameterization and then drawing conclusions from that testing is prone to some circular or flawed logic.

11) The limitation of uncertainty in the lightning data is already included on p.20 line 10 where we see a major problem in conserving the correlation between lightning and other climate variables. We include now that the CAPE anomalies are derived from a global numerical weather prediction model. However, we don't see a flawed logic in showing that although the imposed lightning was strongly increasing the model results don't necessarily show

increases. That the present trend in the imposed lightning leads to a small change in burned area shows that the models have a low sensitivity to lightning. Lightning parameterizations of climate models are tested (see for instance Krause et al. (2014)). Krause et al. (2014) only show a decrease of lightning of 3.3% in pre-industrial times compared to present day. We add this information to give the reader an insight on the uncertainty. The results in Krause et al. (2014) however support our conclusion of the low sensitivity as they also only find small influences on burned area. Using the lightning dataset from Krause et al. (2014) instead of ours would likely reduce the response in burned area.

We add in the manuscript:

Most of the models show a low response of burned area to lightning (see fig. 2), although lightning rates increase by 20% over the simulation period - an increase that is much larger than the 3.3% change between pre-industrial times and the present estimated from a recent modelling study (Krause et al., 2014)

Figure 2 – please retitle these with something that is easier to quickly interpret without cross-referencing the table. For example, I suggest (a) Constant CO2 (SF2\_CO2), (b) Constant Population (SF2\_FPO), (c) Constant Land Cover (SF2\_FLA), (d) Constant Lightning (SF2\_FLI), (e) Constant climate (SF2\_CLI). Also please make figure 2 much wider to avoid the visual clutter of overlaid zigzagging lines. & Figure 2 – change the y-axes ranges so they are constant. It is hard to understand the sensitivity if the plotted range is variable.

12) We changed the Figure according to the suggestions.

Page 11 line 9 – I agree that the statistics suggest individual trends are significant but this does not preclude the massive spread (both positive and negative) in the trends amongst models (table 4). I think this statement needs to include that caveat for an honest accounting of the FireMIP output.

13) The preceding sentence in the manuscript describes the details of the directions of the trends, including positive and negative trends.

Section 3.3 – the first paragraph makes no sense. What I am reading in this study is that the models barely agree on any trend, but yet the authors propose here that the models are important for understanding projected trends and supporting land management strategies. To me, a land management practice cannot be based on model trends that do not agree on trend and cannot be of much use if there is lack of agreement at country scales, let alone finer spatial scales.

14) We agree to some extent, that is why we wrote that the models need to be improved to be useful. We rephrase the paragraph and remove the reference to land management.

Global vegetation models are an important tool for examining the impacts of climate change and are used in policy-relevant contexts (IPCC, 2014; Schellnhuber et al., 2014; IPBES, 2016). Given the various influences of fire on the ecosystems (Bond et al., 2015), the carbon cycle and climate (Lasslop et al., 2019), improvements of global fire models are particularly important.

Section 3.3, second paragraph – the results presented in the manuscript clearly show that models only agree in magnitude in the present day, but the quick microscope analysis of the present day trends show that observations and models do not agree in trends. Some models predict a positive slope, some negative. Unless the authors intend to propose that one FireMIP model is more physically realistic than another, then the results of the sensitivity studies are inconclusive.

15) We agree with the reviewer that we cannot conclude from these analyses how the drivers caused real trends in fire regimes as the divergence between the models is too big. Only a few years ago it was not possible to detect any trends in the satellite data, the satellite estimate is still far from robust. The result of our sensitivity study is an improved understanding of how the trends are caused in the models and how certain trends can be achieved. We have rephrased the paragraph substantially, see reply 1.

Section 3.3 or 4 – it would be useful if these authors were to comment directly on fire models that did not contribute to FireMIP but that have contributed significantly to discussions of human-driven fire both in the present day and over the more distant past. This includes studies by Pfeiffer et al

https://www.geosci-model-dev.net/6/643/2013/, Rabin et al

https://www.geosci-model-dev.net/11/815/2018/, and Hantson et al https://journals.ametsoc.org/doi/full/10.1175/BAMS-D-15-00319.1 . All of these either echo or predict the results discussed by Andela et al 2017 and Bistinas et al 2014 related to a need to quantitatively represent the human use of fire on our planet in the modeling framework.

16) The previous papers acknowledged that the understanding of the human-fire relationship was rather low. However they could not provide the insight that this causes the largest divergence between global fire models as they were not based on a systematic comparison of simulation results. Moreover, we attribute specific model behaviour to the underlying model assumptions. We agree that some of these previous models give important information regarding incorporation of human-fire relationships (but Hantson et al. 2016 only summarizes the discussions of a workshop). Pfeiffer et al. (2013) deal with pre-industrial fire regimes. Rabin et al. (2018) is limited to the period of satellite observations, as they prescribe the agricultural burning based on satellite observations. We integrate these earlier studies in section 3.3 and improve the discussion of the implications for model development. For the full context, see reply 1. Our analysis shows which parts of the models are particularly important to simulate changes in burned area and need additional observational constraints or improved process understanding. In line with previous research (Bistinas et al., 2014; Hantson et al., 2016a, b; Andela et al., 2017), the large divergence in the response to human activities between the FireMIP models shows that the human impact on fires is still insufficiently understood and therefore not constrained in current models.

[...]

Fires on pasturelands have been estimated to contribute over 40% of the global burned area (Rabin et al., 2015). Pasture fires are not treated explicitly in any of the models, although some models slightly modify the vegetation on pastures by harvesting or changing the fuel bulk density (see tab. 5). Expansion of pastures is mostly implemented by simply increasing the area of grasslands. Information on how fuel properties differ between pastures and natural grasslands could therefore help to improve model parametrisations. Prescribing fires on anthropogenic land covers can be a solution for certain applications of fire models (Rabin et al., 2018).

[...]

Regional analysis of remote sensing data could be highly useful, as a global relationship between burned area and individual human factors as assumed in many models and also statistical analysis is not likely. Assumptions on how different human groups (hunter-gatherers, pastoralists, and farmers) use fire have been included in a paleofire model (Pfeiffer et al., 2013). The development of such an approach for modern times would be highly valuable for fire models that aim to model the recent decades and future.

Conclusions – the conclusions are already evident in the Andela et al 2017 paper, so I do not see what we gain in this study. The authors conclude "further analyses are required to better disentangle" factors, but this is the same conclusion so many firemodel and FireMIP papers have arrived at. Could the authors make a clearer argument about what we gain in this manuscript?

17) The cited phrase is not part of our conclusion sections, but part of the discussion. We delete it as it was not a substantial remark. For the gains of the manuscript see reply 1, 9, 16.

**Sensitivity Response** of simulated **historical** burned area to **historical changes in** environmental and anthropogenic **controlsfactors**: A comparison of seven fire models**

Lina Teckentrup1, Sandy P. Harrison2, Stijn Hantson3, Angelika Heil1, Joe R. Melton4, Matthew Forrest5, Fang Li6, Chao Yue7, Almut Arneth3, Thomas Hickler5, Stephen Sitch8, and Gitta Lasslop1,5 1Max Planck Institute for Meteorology, 20146 Hamburg, Germany 2School of Archaeology, Geography and Environmental Sciences (SAGES), University of Reading, Whiteknights, Reading, UK 3Karlsruhe Institute of Technology, Institute of Meteorology and Climate Research, Atmospheric Environmental Research, 82467 Garmisch-Partenkirchen, Germany 4Climate Research Division, Environment Canada, Victoria, BC, V8W 2Y2, Canada 5Senckenberg Biodiversity and Climate Research Institute (BiK-F), 60325 Frankfurt am Main, Germany 6International Center for Climate and Environmental Sciences, Institute of Atmospheric Physics, Chinese Academy of Sciences 7Laboratoire des Sciences du Climat et de l'Environnement–Institute Pierre Simon Laplace, Commissariat à l'Énergie Atomique et aux Énergies Alternatives (CEA)-Centre National de la Recherche Scientifique (CNRS)-Université de Versailles Saint Ouentin 8College of Life and Environmental Sciences, University of Exeter **Correspondence:** Gitta Lasslop (gitta.lasslop@senckenberg.de)

**Abstract.** Understanding how fire regimes change over time is of major importance for understanding their future impact on the Earth system, including society. Large differences in simulated burned area between fire models show that there is substantial uncertainty associated with modelling global change impacts on fire regimes. We draw here on sensitivity simulations made by seven global dynamic vegetation models participating in the Fire Model Intercomparison Project (FireMIP) to understand

5 how differences in models translate into differences in fire regime projections. The sensitivity experiments isolate the impact of the individual drivers of fireon simulated burned area, which are prescribed in the simulations. Specifically these drivers are atmospheric  $CO_2$  concentration, population density, land-use change, lightning and climate.

The seven models capture spatial patterns in burned area. However, they show considerable differences in the burned area trends since 1900.1921. We analyse the trajectories of differences between the sensitivity and reference simulation to improve

10 our understanding of what drives the global trend trend in burned area. Where it is possible, we link the inter-model differences to model assumptions.

Overall, these analyses reveal that the strongest differences leading to diverging trajectories largest uncertainties in simulating global historical burned area are related to the way representation of anthropogenic ignitions and suppression, as well as the and effects of land-use on vegetation and fire, are incorporated in individual models. This points to a. In line with previous studies

15 this highlights the need to improve our understanding and model representation of the relationship between human activities and fire to improve our abilities to model fire for global change within Earth system model applications. Only two models show a

strong response to atmospheric  $CO_2$  and the concentration. The effects of changes in atmospheric  $CO_2$  concentration on fire are complex and quantitative information of how fuel loads and flammability change due to this factor is missing. The response to lightning on global scale is lowfor all models. The sensitivity to climate shows a spatially heterogeneous response and globally only two models show a significant trend. It was not possible to attribute the climate-induced changes. The response of burned

- 5 area to climate is spatially heterogeneous and has a strong interannual variation. Climate is therefore likely more important than the other factors for short term variations and extremes in burned areato model assumptions or specific climatic parameters. However, the strong influence of climate on the inter-annual variability in burned area, shown by all the models, shows that we need to pay attention to the simulation of fire weather but also meteorological influences on biomass accumulation and fuel properties in order to better capture extremes in fire behavior. This study provides a basis to understand the uncertainties
- 10 in global fire modelling and the necessary improvements in process understanding and observational constraints to reduce uncertainties in modelling burned area trends.

Copyright statement. TEXT

**1 Introduction**

About 4of the global vegetated area burns each year (Giglio et al., 2013), but between 30-40of the land surface is affected

- 15 by fire regularly (Chapin et al., 2002; Chuvieco et al., 2008). Thus, over large parts of the world, wildfires Wildfires are an important cause of vegetation disturbancedriver of vegetation distribution, and regulate ecosystem functioning, biodiversity and carbon storage over large parts of the world (Bond et al., 2005; Hantson et al., 2016a). Fire has strong impacts on climate through changing land surface properties, atmospheric chemistry and hence radiative forcing, as well as biogeochemical cycling (Bowman et al., 2009; Randerson et al., 2012; Ward et al., 2012; Yue et al., 2016; Li and Lawrence, 2017; Li et al., 2017)
- 20 (Bowman et al., 2009; Randerson et al., 2012; Ward et al., 2012; Yue et al., 2016; Li and Lawrence, 2017; Li et al., 2017; Lasslop et al., 2 Estimates of the net effect of fire on the Earth system vary. Analyses based on observations of the pre-industrial period suggest that the contribution of fire to the overall climate–carbon-cycle feedback is substantial (with 5.6 ± 3.2 ppm K-1 CO2 per degree of land temperature change; Harrison et al., 2018) (Harrison et al., 2018) while the strength of the global land climate–carbon-cycle feedback estimated from Earth system simulations (Arora et al., 2013) is 17.5 ppm K-1
- 25 (Harrison et al., 2018). However, comparing potential fire-induced losses from terrestrial carbon pools and stocks of solid pyrogenic carbon in soils and ocean, fire may also be a net sink of carbon and Earth system simulations show a negative effect of fire on radiative forcing (Lasslop et al., 2019). In addition to these consequences for the Earth System, wildfires directly impact society and economy (Gauthier et al., 2015) and human health can be seriously impaired (Johnston et al., 2012; Finlay et al., 2012).
- 30 Given the various impacts that fire has of fire on natural and human systems and the large uncertainties, it is important to understand-improve the understanding on what controls the occurrence of wildfires and to know how fire regimes might

change in the future.

The fire model intercomparison project (FireMIP, Hantson et al., 2016a; Rabin et al., 2017a ) provides simulations for a systematic comparison of fire-model behaviour based on outputs from a large range of multiple model runs with identical forcing inputs. Sensitivity simulations were conducted for individual forcings, specifically CO2, population density, land-use change, lightning

and climate.Based on current process understanding these drivers may influence burned area in the following waysthe following drivers influenced burned area over the last decades to centuries:
 Increasing atmospheric CO2 concentration leads to increases in net primary production (Hickler et al., 2008; Knorr et al., 2016) (Hickler et al., 2008) and decreased stomatal conductance reduces the plant transpiration and enhances water conservation in

plants (Morison, 1985). It can lead to changes in vegetation composition an increase in the abundance of woody plants ('woody

- 10 thickening'; Wigley et al., 2010; Bond and Midgley, 2012; Buitenwerf et al., 2012) because  $C_3$  plants are generally more competitive than  $C_4$  plants under higher atmospheric CO2 concentration (e.g. Ehleringer and Björkman, 1977; Ehleringer et al., 1997; Wand et al., 2001; Sage and Kubien, 2007). The impact of these various changes on burned area is complex. Increased productivity can lead to increased fuel availability, which can lead to increased burned area in water- and fuel-limited regions (Kelley and Harrison, 2014). On the other hand, decreased stomatal conductance and lower transpiration can lead to increased
- 15 soil and enhanced water conservation in plants. This increases the moisture content of soil as well as vegetation moisture content and consequently live and dead fuel moisture contentsand hence to a reduction in burned area in more humid regions, which decreases flammability and in consequence reduces burned area. Woody thickening can lead to a reduction in burned area through changing the nature of fuel loads (Kelley and Harrison, 2014).
- There is still controversy about whether humans increase or decrease fire overall: Although there is broad agreement that hu-20 mans suppress fires in regions with high population density, observational studies are less clear about what happens in areas of low population density and show both increases or decrease decreases due to human activities (see for instance Marlon et al., 2008; Bowman et al., 2011; Marlon et al., 2013; Vannière et al., 2016; Andela et al., 2017; Balch et al., 2017). Studies of the covariation between population density and number of fires have shown that increasing population density leads to an increase in the number of ignitions or in the number of individual fires until peaking at inter-mediate-intermediate population
- 25 densities and drop subsequently (Syphard et al., 2009; Archibald et al., 2010). Burned area can be expressed as the number of fires multiplied by their fire size. The increase in burned area for low population density due to changes in ignitions is expected to differ from the one found for number of fires between regions with varying population density as the largest fires occur in unpopulated areas (Hantson et al., 2015a)and burned area can be expressed as number of fires times fire size. Global analysis . Global analyses find that the net effect of population density is a decrease in burned area (Bistinas et al., 2014; Knorr et al., 2015; Knorr et al., 2014; Knorr et al., 2014; Knorr et al., 2014; Knorr et al., 2015; Knorr et al., 2014; Knorr et al., 2014; Knorr et al., 2014; Knorr et al., 2015; Knorr et al
- 2014), with high uncertainties for low population density if the method allows for non-monotonic relationships (Knorr et al., 2014). Regional analysis tends analyses tend to confirm this, but positive relationships between burned area and population density have been shown, for instance, for the least disturbed areas in the USA (Parisien et al., 2016).
   First uncertainties analysis in the inductrial times (a p. Durnend, 10(1), Otto, and Anderson, 1092). Jakastan, 2002)

Fire was used to manage croplands in pre-industrial times (e.g. Dumond, 1961; Otto and Anderson, 1982; Johnston, 2003) and it-is still common practice in-mainly in non-industrialized areas (i.e. Sub-Saharan Africa, parts of South East Asia, In-35 donesia and Latin America; e.g. Conklin, 1961; Rasul and Thapa, 2003). However fires in agricultural areas are common on

3

all over the world (Korontzi et al., 2006). The influence of land-use on fire on global scale is not well studied. Severe data gaps and an unsatisfactory level of understanding characterize our knowledge 
[revised manuscript text omitted]

- 30 behavior during the 20th century, this difference in the length of the simulations between the models should have little impact and due to limitations in data availability the reference year of the forcings used in the spin-up varies (see tab. 1). We account for these differences in starting years between models and of the forcing factors by limiting our ana

---

## Author Response (AR2)

Dear Editor,

we reply below to the second review. Our responses are formatted in italics and suggested new text is highlighted in red.

1. in their replies, the Authors emphasize the uncertainty in satellite burned area products, but while i agree that there is uncertainty, the Authors may want to emphasize what improvements in satellite records are necessary to be an effective tool in evaluating fire models. i guess i am unclear whether any improvement in satellite records would suffice given these following confusing statements:

1) *We thank the reviewer for pointing out these confusing sentences. We reply to each of the comments below. But please note also that we already had included a paragraph to explain how recent developments in satellite products could help to improve fire models:*

   *Recent advances in remote sensing products have high potential to support model development. However, remotely sensed burned area datasets alone are not a sufficient basis to evaluate fire models as many model structures can lead to reasonable burned area patterns. The emergence of longer records of burned area and the increasing availability of information on other aspects of the fire regime considerably improve opportunities to evaluate and improve our models. The FRY database (Laurent et al., 2018) and the global fire atlas (Andela et al., 2018), for example provide information on fire size, numbers of fire, rate of spread, and the characteristics of fire patches. These datasets will be useful to, for instance, separate effects of ignition and suppression. Rate of spread equations in global fire models are at present either very simple empirical representations tuned to improve burned area or based on laboratory experiments (Hantson et al., 2016). The mentioned datasets now offer the opportunity to derive parameters for rate of spread equations at the spatial scales these models operate on. Fire size and rate of spread are important target variables besides burned area that can determine the impacts of fire. The effects on vegetation (combustion of biomass and tree mortality; Williams et al., 1999; Wooster et al., 2005) and on the atmosphere (Veira et al., 2016) are a function of fire intensity, which is also included in the FRY database (Laurent et al., 2018). A better evaluation of such parameters can enhance the usability of fire model simulations.*

Reply 1: "The spatial patterns of burned area and its drivers are already well explored and understood." (Comment: isn't there uncertainty in the drivers related to humans/population and lightning? maybe the Authors need to be more specific in what exactly is understood)

   2) *We agree that there is always uncertainty and the wording did not really get to the point. We now reword it to:*

   *The information content on the spatial variability of burned area has been well exploited in previous studies and models reproduce the spatial patterns in a reasonable way. The temporal information of the satellite data is increasing with the increasing length of the record and has a higher potential to contain new information to support the improvement and evaluation of global fire models.*

Reply 1: "However, remotely sensed burned area datasets alone are not a sufficient basis to evaluate fire models as many model structures can lead to reasonable burned area patterns." (Comment: what kind of satellite burned area product would work then to evaluate fire models? insights from the Authors on that point may help guide remote sensing data products. also, doesn't this contradict the Reply 1 statement "The spatial patterns of burned area and its drivers are already well explored and understood."?)

   3) *We go more into detail about which datasets can be helpful to improve models in the suggested text changes following directly the quoted sentence. See 1) .*

Reply 4: "Moreover we now include a new publication (Forkel et al. 2019b) in the discussion which shows that the trends as observed by satellites are still highly uncertain and not robust." (Comment: what does highly uncertain mean? part of this is that i don't know the Forkel et al 2019b paper, but Authors could be more specific)

   4) *We already included specific information in on the uncertainties in the suggested changes in the reply and in the revised manuscript:*

   *"However, as Forkel et al. (2019b) have shown, the significance of the observed global decline is strongly affected by the length of the sampled*

*interval because of the high interannual variability in burned area and trends between products show only a low correlation (Forkel et al., 2019b)."*

2. fairly simple is that the Authors forgot to provide a citation or DOI for the Forkel et al. 2019b paper which i would like to see since it is relevant to point 1.

5) *We apologize for not including the reference in the replies. It was only included in the revised manuscript and the tracked changes version. The reference is: 
[revised manuscript text omitted]